# Template switching between the leading and lagging strands at replication forks generates inverted copy number variants through hairpin-capped extrachromosomal DNA

Rebecca Martin ‡, Claudia Y. Espinoza ‡, Christopher R. L. Large, Joshua Rosswork, Cole Van Bruinisse, Aaron W. Miller, Joseph C. Sanchez, Madison Miller, Samantha Paskvan, Gina M. Alvino, Maitreya J. Dunham, M. K. Raghuraman, Bonita J. Brewer *

Department of Genome Sciences, University of Washington, Seattle, Washington, United States of America

‡ These authors share first authorship on this work.
* bbrewer@uw.edu

**Data Availability Statement:** All data are included as supplemental tables or deposited in the NIH

## Abstract

Inherited and germ-line *de novo* copy number variants (CNVs) are increasingly found to be correlated with human developmental and cancerous phenotypes. Several models for template switching during replication have been proposed to explain the generation of these gross chromosomal rearrangements. We proposed a model of template switching (ODIRA—origin dependent inverted repeat amplification) in which simultaneous ligation of the leading and lagging strands at diverging replication forks could generate segmental inverted triplications through an extrachromosomal inverted circular intermediate. Here, we created a genetic assay using split-*ura3* cassettes to trap the proposed inverted intermediate. However, instead of recovering circular inverted intermediates, we found inverted linear chromosomal fragments ending in native telomeres—suggesting that a template switch had occurred at the centromere-proximal fork of a replication bubble. As telomeric inverted hairpin fragments can also be created through double strand breaks we tested whether replication errors or repair of double stranded DNA breaks were the most likely initiating event. The results from CRISPR/Cas9 cleavage experiments and growth in the replication inhibitor hydroxyurea indicate that it is a replication error, not a double stranded break that creates the inverted junctions. Since inverted amplicons of the *SUL1* gene occur during long-term growth in sulfate-limited chemostats, we sequenced evolved populations to look for evidence of linear intermediates formed by an error in replication. All of the data are compatible with a two-step version of the ODIRA model in which sequential template switching at short inverted repeats between the leading and lagging strands at a replication fork, followed by integration via homologous recombination, generates inverted interstitial triplications.

Sequence Read Archive (SRA) under BioProject ID PRJNA1016460.

**Funding:** This project was supported by National Institutes of Health (https://www.nih.gov) grants R01 GM018926 and R35 GM122497 to BJB and MKR; and National Science Foundation (https://www.nsf.gov) grant 1120425 and National Institutes of Health (https://www.nih.gov) grant P41 GM103533 to MJD. AWM and CRLL were supported in part by National Institutes of Health (https://www.nih.gov) grant T32 HG00035. The funders had no role in study design, data collection and analysis, decision to publish, or preparation of the manuscript.

**Competing interests:** The authors declare that they have no competing interests.

## Author summary

Chromosomal rearrangements are a potent source of genetic variation in humans and other organisms. One specific type of rearrangement involves the increase in copies of segments of the genome. The variation in gene dosage that these rearrangements can cause has been associated with a wide range of neurological and other human disorders. A specific puzzling form of copy number increase consists of three tandem copies with the central copy in inverted orientation. How this rearrangement occurs is of great interest, yet the mechanisms responsible are only inferred by examining the sequence of final inverted products. Yeast provides a unique model system to explore the underlying molecular defects that give rise to inverted triplications. While the favored hypothesis suggests that double stranded DNA repair is the causative agent, we find that a particular form of template switching between strands at the replication fork, not a double stranded DNA break, is the initiating event. Using the awesome power of yeast genetics, we provide evidence in two different assays for this unique replication error that we call ODIRA (for Origin Dependent Inverted Repeat Amplification) and propose that it can also explain this form of copy number variant seen in human evolution and disease.

## Introduction

Copy number variation (CNV) refers to both increases and decreases in copies of genomic segments. In humans, many CNVs not only distinguish us from our close primate relatives, but some arise *de novo* and are associated with a range of human disorders [1–5]. One of the most common forms of CNV found in the human genome is the repetition of large genomic segments (referred to collectively as segmental duplications). Although the extra copies can be found as a direct repeat at the original locus, they may also be found at dispersed sites on the same or different chromosomes [6]. There are three major pathways that are thought to give rise to changes in copy number through distant interactions: non-allelic homologous recombination (NAHR), non-homologous end joining (NHEJ) and template switching during replication (FoSTeS and MMBIR) (Reviews: [7–13]). However, these mechanisms fail to easily and/or completely explain the formation of an unusual form of segmental duplication that involves a tandem triplication, where the central copy is inverted within an otherwise unrearranged chromosome. This form of CNV is seen in many human disorders and is likely under-reported because determining its inverted structure is technically challenging [8,14,15].

During long-term growth of the common laboratory strains of haploid budding yeast *Saccharomyces cerevisiae* in chemostats limiting for sulfate, we routinely recover inverted triplications of the *SUL1* locus, which encodes the primary sulfate transporter [16,17]. The reproducibility of this outcome provides an ideal system and opportunity to investigate the mechanism that gives rise to this form of gene amplification. Although the size of the amplified region varies, several structural features appear to be invariant (Fig 1A.1): (1) the amplified segment contains at least one origin of replication, (2) the junctions that mark the boundaries of the amplified segment occur at pre-existing short, interrupted inverted repeats, and (3) the arms of the inverted repeats used for amplification are within a hundred base pairs of each other [16–18]. We have proposed a unique template-switching model, called ODIRA (Origin-Dependent Inverted Repeat Amplification; [19]), in which the leading strands at divergent, stalled replication forks become ligated to the Okazaki fragments on the lagging strands due to strand migration (Fig 1A.2; "dogbone" ODIRA). Displacement and replication of the closed loop of newly synthesized DNA—the dogbone—gives rise to an inverted, dimeric, circular

**Fig 1. Comparison of "dogbone" and "hairpin" ODIRA models for inverted triplication of the *SUL1* locus.** In the following diagrams thick lines indicate double stranded duplexes and thin lines indicate individual single strands. A) In our ODIRA model we propose that stalled forks (2a) provide an opportunity for a template switch between the nascent leading strand and the lagging strand template that occurs at short, interrupted inverted repeats (2b). Extension of the displaced leading strand and its ligation to an Okazaki fragment (2c) results in a covalent linkage between the leading and lagging nascent strands that can be expelled from the chromosome by an incoming fork from an adjacent origin (2d, and 3). A similar template switch at the divergent fork results in an extrachromosomal, self-complementary, single-stranded circular molecule (dogbone; 3). In the next cell cycle, the dogbone can replicate from its resident origin creating a duplex circular molecule that has two copies of the *SUL1* region in inverted orientation (4). Recombination of the inverted dimeric circular molecule into the chromosome through homology with the *SUL1* region creates a triplication with the center copy in inverted orientation (5). The inversion junctions (Cen-proximal and Tel-proximal; CJ and TJ, respectively) map to the genomic, short interrupted, inverted repeats where the template switching occurred. B) The Cen-proximal and Tel-proximal inversions can occur in different cell cycles, generating inverted linear molecules. After the second, telomere-proximal junction is created, the doubly inverted linear molecule can recombine with the *SUL1* region creating an inverted triplication that is identical to that produced by the dogbone ODIRA model. The gray shaded panels in A and B show both strands of the DNA as thin lines to highlight the mechanism of template switches and the expelled transient intermediates (dogbones and hairpins). The dotted rectangles indicate the final chromosomal products of the two pathways.

DNA molecule containing *SUL1* and the adjacent origin of replication (Fig 1A.3-4). (We use the term dogbone to refer specifically to the expelled closed DNA with single stranded loops. After replication we refer to the double stranded product as an inverted dimeric circular molecule.) Subsequent integration of the inverted dimeric circle into the chromosome at the original locus generates the triplication with an inverted center copy without disturbing the distal chromosomal sequences (Fig 1A.5).

The amplicon can also arise in a two-step ODIRA mechanism (Fig 1B.1; "hairpin" ODIRA) where the two template switches are temporally uncoupled. Ligation of leading and lagging strands at just the single fork moving toward the centromere (centromere-proximal junction; CJ) could produce an intermediate consisting of a hairpin capped linear segment extending to the telomere that could persist as an extrachromosomal inverted linear duplex after replication (Fig 1B.2-3). (We use the term hairpin to refer specifically to the expelled double stranded linear with a single stranded loop at one end. After replication we refer to the completely double stranded product as an inverted linear molecule.) Recombination with the initiating chromosome does not result in integration, but rather just shuffles the telomeric arms between the inverted linear and the chromosome. However, cells containing the inverted linear would enjoy a selective advantage in the sulfate-limited chemostat. In a subsequent cell cycle, a second leading-to-lagging strand template switch in the fork moving toward the telomere (telomere-proximal junction; TJ; Fig 1B.4-6) of the inverted linear would generate a doubly inverted linear that could recombine with the *SUL1* chromosome and generate the inverted triplication, improving its stability and selective advantage that leads to this clone sweeping the population. Notice that recombination with either of the internal repeats produces the triplication while recombination with either of the more distal repeats just shuffles the telomeres.

Two other models of template switching—FoSTeS and MMBIR—have been proposed to account for complex chromosomal rearrangements involving distant interactions [9,10]. These models involve the migration of the 3' end of a nascent strand from a replication fork or the exposed 3'end from a double stranded break to other regions of homology elsewhere in the genome. At the new site, replication is reestablished, generating a junction between two disparate regions of the genome. To explain complex rearrangements, the model proposes that the same strand makes multiple sequential invasion/extension attempts in a single cell cycle. Inverted triplications would not require long-distance template switching since the homologous template is the opposite strand at the replication fork (Fig 1A.2 and 1B.1).

The nature of the inverted junctions inspired the ODIRA model. One well characterized example from the human literature seemed to fit perfectly with ODIRA because the triplication that occurred in the father of the female proband was a 2:1 mixture of SNPs from his two homologues [20]. An inverted dimeric circle that arose from one homologue and inserted into the other homologue during or preceding meiosis could be the explanation for the 2:1 SNP ratio. This case provided us with the stimulus to artificially recreate such an event in yeast by asking whether we could detect the movement of the inverted dimeric circle (produced by template switching across the replication fork) to a new location in the yeast genome. We designed a split-*ura3* construct to identify the movement of potential extrachromosomal intermediates from one location to another by the recreation of a functional *URA3* gene from two overlapping partial *ura3* fragments.

To capture potential inverted extrachromosomal intermediates, we integrated a 5' fragment of *URA3* ("*ura*") on chromosome II at the *SUL1* locus, a region that is prone to inverted triplications in sulfate-limited cultures. On chromosome IX we integrated the overlapping partially complementary "*ra3*" fragment. Direct recombination between the homologous regions on chromosome II and IX would produce unstable dicentric chromosomes; however, an ODIRA event—a circular inverted dimeric *ura* fragment integrating into chromosome IX—would

recreate a functional *URA3* gene along with a partial inverted triplication on chromosome IX. Among the hundreds of clones analyzed we failed to detect any Ura+ clones that were consistent with integration of **circular** intermediates into chromosome IX. Instead, we identified Ura+ clones with genomic rearrangements that could be explained by a recombination between an inverted **linear** *ura* fragment from chromosome II with the *ra3* fragment on chromosome IX (ODIRA hairpin model).

Creation of identical linear extrachromosomal intermediates could be explained by repair of a double stranded DNA (dsDNA) break that forms intrastrand hairpins at short, interrupted palindromes near the break. To distinguish between these two mechanisms we carried out the selection for Ura+ clones under two conditions: (1) altering the strand dynamics at replication forks by reducing the availability of dNTPs with two different concentrations of hydroxyurea, and (2) increasing dsDNA breaks by targeting CRISPR/Cas9 centromere proximal to the *SUL1* locus. These experiments suggest that template switching between the leading and lagging strands at a replication fork, but not double stranded DNA breaks, initiates the production of inverted amplicons.

To determine whether the hairpin ODIRA model could explain the occurrence of inverted triplication of the *SUL1* locus in sulfate-limited chemostats [16–18], we sequenced 31 independent populations of cells that had been passaged for ~250 generations. By focusing on split reads that define junctions of inversions, we discovered that nearly a third of the cultures had different numbers of Cen-proximal and Tel-proximal inverted junctions, suggesting that there were inverted linear segments produced during the experiments. Moreover, the spacing of short-inverted repeats, the orientation of the inverted junctions and the positions of the junctions with respect to the replication map of this region of chromosome II provide further evidence that template switching between strands at a replication fork, not double strand DNA breakage, initiates inverted gene amplification in yeast.

## Materials and methods

### Yeast strains and culture conditions

BY4741 (*MATa, his3Δ1 leu2Δ0 met15Δ0 ura3Δ0*) and BY4742 (*MATα, his3Δ1 leu2Δ0 lys2Δ0 ura3Δ0*) were used to construct the split-*ura3* strain. The two partially overlapping regions of the *URA3* gene, referred to as *ura* and *ra3*, were generated in two steps. We first selected (on -uracil plates) for *URA3* insertion on chromosome II and chromosome IX, respectively into BY4741 and BY4742, by transformation with a PCR fragment derived from pRS406 with 100 bp homology arms (S1 Table; SUL1_URA3_F and SUL1_URA3_R; Chr9_URA3_Chr9_F and Chr9_URA3_Chr9_R). These strains were then transformed with truncated PCR fragments of *ura* and *ra3*, similarly created from pRS406 and primers with the same homology arms (S1 Table; SUL_URA3_F and ura_SUL1_R; Chr9_ra3_F and Chr9_URA3_Chr9_R). Transformants were selected on plates with 5-fluoro orotic acid (5-FOA) and confirmed by PCR/ Sanger sequencing and Southern blotting of CHEF gels. BY4741 containing the *ura* fragment was mated to BY4742 containing the *ra3* fragment to create the doubly heterozygous diploid. Sporulation and tetrad dissection resulted in a haploid spore, s2-1 (*MATa, ura3Δ, his3Δ1, leu2Δ0, lys2Δ0, sul1::ura, FAT1-3'::ra3*; hereafter referred to as the "split-*ura3* strain"; Fig 2A), with both partial *ura3* fragments that were confirmed by CHEF gel/Southern blotting. This strain was used in all subsequent experiments involving selection for Ura+ clones. The *ura* insert lies within the *SUL1* gene between coordinates 789418 and 791405 on chromosome II on the Watson strand. The *ra3* fragment lies in an intergenic region between *FAT1* and *CST26* at coordinates 321188–321194 on chromosome IX, also on the Watson strand ~34 kb upstream of *CEN9* (355629–355745). The overlap between *ura* and *ra3* is 203 bp.

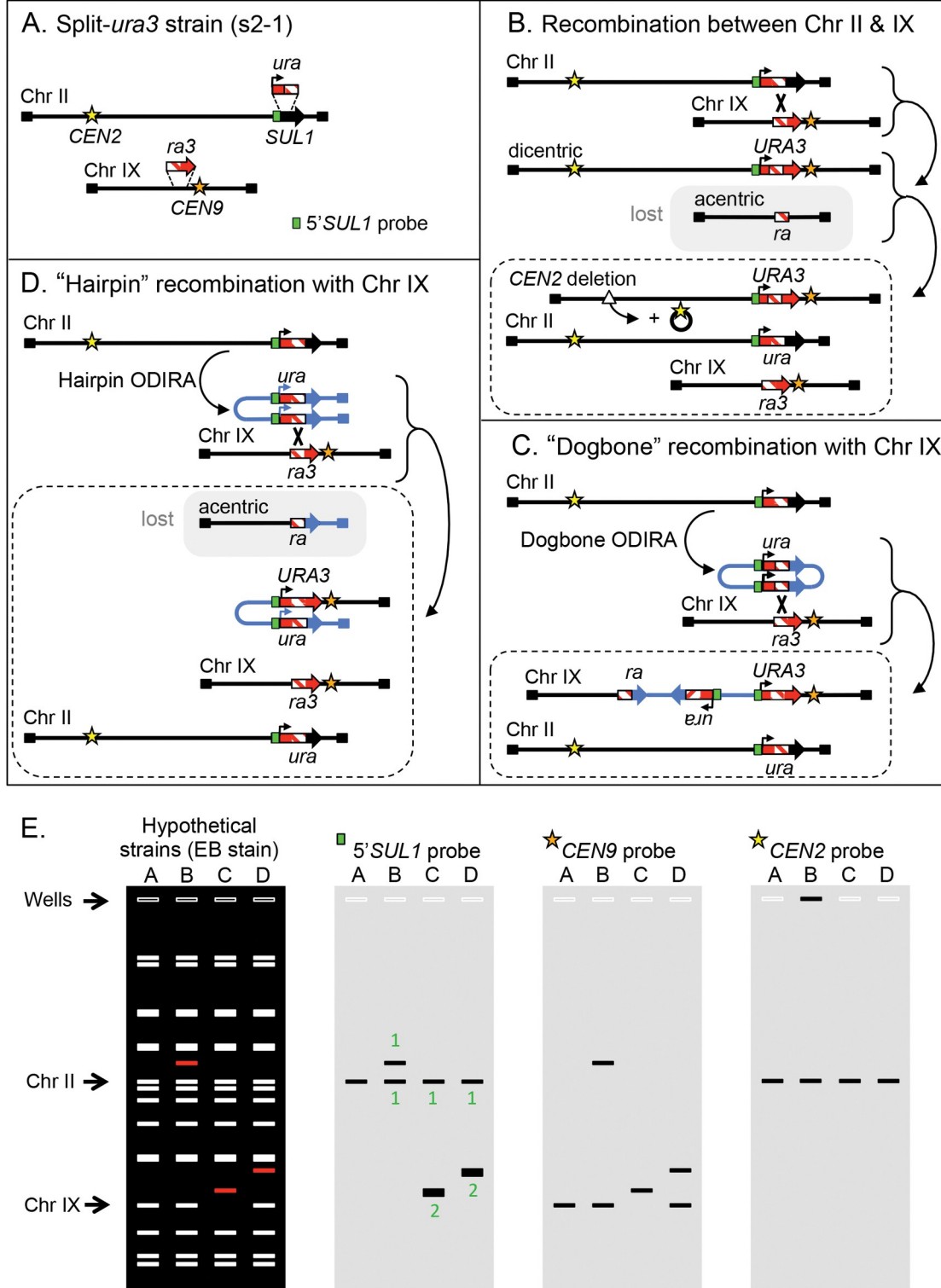

**Fig 2. Experimental design for detecting extrachromosomal intermediates during DNA amplification.** In the following diagrams black lines are used to indicate the original chromosomal sequences and blue lines refer to the ODIRA generated intermediates and their fate after recombination with chromosomes. (A) Sites on chromosomes II and IX were modified by insertion of overlapping fragments of the *URA3* gene. A probe to the unique 5' portion of the *SUL1* fragment used for Southern blotting is highlighted in cyan. (B) Recombination between the two marked chromosomes can recreate an intact *URA3* gene but results in the creation of an unstable dicentric Ura+ chromosome. Deletion of one of the centromeres results in stabilized

chromosome. The reciprocal product is an acentric chromosome that is lost during mitosis. (C) The chromosome II *ura* fragment, amplified as an extrachromosomal, inverted circular molecule, can recombine with the target site (*ra3*) on chromosome IX to generate a functional *URA3* gene. The resulting copy of chromosome IX contains a tandem inverted triplication of the *ra* segment. (D) A single replication error at the centromere-proximal fork generates a palindromic linear fragment. Recombination between one of the copies of the *ura* fragments and the target *ra3* fragment on chromosome IX re-forms a functional *URA3* gene and creates a translocation between the palindromic chromosome and chromosome IX. To cover the loss of essential genes on the left arm of chromosome IX, an unrearranged chromosome IX is also expected. (E) Anticipated CHEF gel results for the four strains described in A-D. The ethidium bromide stained gel reveals all chromosomes—the rearranged chromosome for each case is indicated in red. Hybridization with chromosome specific probes is used to distinguish the different outcomes, including the 5'*SUL1* probe shown in (A). The relative intensities of the 5'*SUL1* probe are indicated for each band as either 1 or 2 (green type). The deleted *CEN2* (B) is often retained as a large circular molecule through recombination between flanking TY repeated elements. In CHEF gels, these circular molecules are often found retained in the well. Dotted rectangles in B, C, D show the expected final products.

FY4 (*MATa*, *srd1Δ*0) was grown in mini-chemostats limiting for either sulfate (31 independent chemostats) or glucose (32 independent chemostats) for ~250 generations [21,22].

## Selection for Ura+ clones

Small independent colonies of the split-*ura*3 strain were picked from a synthetic complete plate with 5-FOA and inoculated in 1 ml of complete synthetic yeast medium and grown to a saturation density of ~$10^8$ cells. The 1 ml of cells was concentrated by centrifugation, plated onto a single -uracil plate, and incubated at 30°C for 3–5 days. A maximum of two colonies from each -uracil plate were restreaked on a fresh -uracil plate. Each purified clone was grown in 8 ml of -uracil liquid medium to make freezer stocks, CHEF-gel plugs and "NIB-n-grab" DNA preps (https://fangman-brewer.genetics.washington.edu/nib-n-grab.html; see below). To test for the effect of nucleotide depletion on the formation of Ura+ clones, the 1 ml of medium contained either 50 or 200 mM HU.

## Contour-clamped Homogeneous Electric Field (CHEF) gel electrophoresis/ Southern blotting

Agarose plugs for CHEF gel electrophoresis were generated using the method by L. Argueso (described in [23]) or by an adaptation of the method by S. Iadonato and A. Gnirke (described in [24]). Run conditions in the BioRad CHEF-DRII were 0.8% LE agarose in 0.5XTBE in 2.3 L 0.5% TBE running buffer at 14°C. Switch times were 47" to 170" at 165V for 39–62 hours. Standard conditions for Southern blotting and hybridization with $^{32}$P-labeled PCR probes are described by Tsuchiyama et al. [24]. Hybridization intensity was determined using a BioRad Personal Molecular Imager. Primer pairs used to create $^{32}$P-labeled PCR probes are given in S2 Table.

## CRISPR/Cas9 cutting in the vicinity of *SUL1*

The split-*ura*3 strain was transformed with plasmid pYCpGal that include a yeast centromere, the yeast *LEU2* gene, the *GAL1* promoter driving *Cas9* expression, and a guide RNA cloning site. The Cas9-guide cassette was derived from plasmid pML104 (Addgene) (S1 Fig). Guide RNAs expressed from this plasmid directed cutting to either position 708.260 kb (pYCpGAL-708b) or 792.883 kb (pYCpGAL-792b) on chromosome II (S3 Table). Relative to *SUL1*, the sites are centromere-proximal and centromere-distal, respectively. Standard LiAc transformation was used to introduce a no-guide plasmid, the 708 plasmid or the 792 plasmid into the split-*ura*3 strain, selecting for transformants on -leucine, +glucose plates. Single colonies were then used to inoculate 1 mL of –leucine +glucose medium. The 1 ml of cells was concentrated and plated on -uracil, -leucine, +raffinose, +galactose to induce cutting by CRISPR/Cas9 and

to select for uracil prototrophy. After restreaking colonies on–uracil plates, clones were expanded in liquid –uracil medium for freezer stocks and CHEF-gel plugs.

## PCR/Sanger Sequencing

The insertions in the split-*ura3* strain were confirmed by PCR and Sanger sequencing, in addition to Southern blotting. The state of CRISPR/Cas9 cutting at the 708 site or the 792 site was confirmed by PCR. Sanger Sequencing of PCR fragments was performed by GeneWiz (Azenta) or Eurofins.

## aCGH (array Comparative Genome Hybridization) analysis

Genomic DNA from frozen chemostat samples from generation ~250 was isolated by the NIB-n-Grab protocol (see above), a modified version of the Smash-and-Grab protocol [25] that results in the recovery of DNA 20–50 kb in size. In this protocol, cells are broken by vortexing with glass beads in a buffer that stabilizes nuclei (NIB; [26]). aCGH was performed using Agilent 4x44k microarrays with probes spaced every 290 nt on average. Hybridization was executed as described previously [21]; however, sonication of the DNA samples was performed after *in vitro* labeling, rather than before labeling. This alteration allows inverted junctions to be identified by the gradual increase in signal—from single copy to multiple copies—at the site of inversion (see S2 Fig). aCGH data for the relevant chromosomes are available in S4 and S5 Tables.

Note: Interstitial inverted triplications in the human CNV literature are referred to as a "triplicated segment embedded in an inverted orientation between two duplicated sequences (DUP-TRP/INV-DUP)" [27]. They are found associated with a variety of genetic syndromes but remain an underappreciated form of CNV, primarily because the inverted nature of the amplicon junctions poses a challenge for DNA sequencing platforms [15]. While arrayCGH has largely been replaced by long read sequencing in genomic research there are inherent problems with nanopore sequencing of inverted templates [28]. We suggest that a modified protocol of aCGH can easily detect inverted boundaries of amplified regions.

## Population short read sequencing

150 bp paired-end sequence of whole genome fragments purified from population samples on the last day of the chemostat run were prepared and analyzed as described [17]. Median genome read depth for the 31 sulfate-limited chemostats was ~125. Read depth analysis for the sulfate-limited cultures showed amplification of the *SUL1* gene and flanking sequences while none of the 32 glucose-limited chemostats had amplifications in the *SUL1* region. All sequencing data are available in the NIH Sequence Read Archive (SRA) under BioProject ID PRJNA1016460.

## Split read analysis

Split reads are defined as 150 bp reads that map to two non-contiguous regions of the yeast genome [17] and were manually curated on IGV (Integrative Genomics Viewer; [29]) as falling into one of five categories: (1) inverted junctions have reads divided between two sections of chromosome II on opposite DNA strands (n = 92); (2) direct repeat junctions have the two segments of the sequence from the same strand of chromosome II (n = 4;); (3) de novo telomere additions contain part of a chromosome II sequence adjacent to a $C_{1-3}A/G_{1-3}T$ telomere sequence (n = 0); (4) telomere translocation junctions have the terminal *SUL1* part of chromosome II-R attached to an existing telomere (n = 6); and (5) internal translocation junctions

join chromosome II sequence to a repetitive element elsewhere in the genome—such as Ty elements, solo delta elements, tRNAs, and polyA/polyT or CAG stretches (unquantified). Inverted junction sequences (category 1) were catalogued (S6 Table) and characterized by the size of the inverted sequence at the discontinuity, the spacing between the two inverted sequences, and orientation of the inversion event. To establish baselines for all available inverted repeats we used the EMBOSS Palindrome program (https://www.bioinformatics.nl/cgi-bin/emboss/palindrome) to find all potential inverted repeats and their spacing across the terminal ~80 kb of Chromosome II with an interrupted inverted repeat structure of ≤250 bp using the sacCer3 version of the genome.

For an inverted split-read junction to be considered significant, we required that at least two independent chromosome fragments from the same culture produce the same junction sequence. Inverted junctions represented by a single read were considered to be PCR artifacts produced during the sequencing protocol, while those represented by more than one unique read were copies of *in vivo* generated inversions.

## Statistical analysis

The significance of differences in the frequency of inverted *URA3* amplicons in normal medium vs. medium containing hydroxyurea (HU, 50 or 200 mM) was assessed by Chi-squared analysis. The significance in inverted repeat length and orientation of inverted junctions among split reads from whole genome sequencing of population samples from sulfate-limiting chemostats was performed using the Mann-Whitney Rank Sum test.

## Results

### Genetic selection and identification of stabilized ODIRA-generated intermediates

One key feature of both versions of the ODIRA model relies on the presence of extrachromosomal intermediates that reintegrate into the chromosome through homology-dependent recombination to give rise to the inverted amplicons. To determine whether such intermediates exist, we designed a system in which homology for the integration event is at a new site where it generates a selectable phenotype. We constructed the haploid strain s2-1 (hereafter referred to as "the split-*ura3* strain") in which partially overlapping *URA3* gene fragments, with a shared central region of identity, were integrated on two different chromosomes (Fig 2A). We replaced a section of the *SUL1* gene on chromosome II with the 5' portion of the *URA3* gene (*ura*) in the same transcriptional orientation as *SUL1*. We inserted the overlapping 3' portion (*ra3*) in a non-essential site on chromosome IX to the left of *CEN9* with a transcriptional orientation toward *CEN9* (Fig 2A). The design of our system allows for the reconstitution of the *URA3* gene through at least two different mechanisms. One way involves integration of intermediates from either of the two ODIRA models. The inverted linear or the inverted dimeric circle (i.e., the replicated forms of linear or dogbone intermediates, respectively), carrying the *ura* sequence, can recombine with the *ra3* sequence on chromosome IX to generate Ura+ prototrophs. A second way of reconstituting the *URA3* gene is through a direct recombination event between the two chromosomes within the region of *ra* homology. This reconstitution, however, gives rise to an unstable dicentric chromosome and a reciprocal acentric chromosome that is lost in subsequent cell divisions (Fig 2B). Our experimental set-up permits us to detect the insertion of the inverted dimeric circular intermediate onto chromosome IX as well as a recombination event with the inverted palindromic linear (Fig 2C and 2D). Both recombination events would be mitotically stable and their chromosome structures

would be easily distinguishable by CHEF gel electrophoresis and Southern blotting from each other and from events that resulted from direct recombination (Fig 2E).

Starting with individual colonies of the split-*ura3* strain grown on 5-FOA (to ensure that they were phenotypically uracil auxotrophs), we grew cultures to stationary phase in 1 ml of liquid complete medium before plating ~$10^8$ cells on plates lacking uracil (-uracil plate). Each expanded colony generated at least one Ura+ clone. In the initial experiments we chose one or two colonies from each -uracil plate (27 unique events) and analyzed their chromosomes by CHEF gels and Southern blots (Fig 3A). If recombination between the two chromosomes generated the *URA3* gene, then a single copy of the 5'*SUL1* sequence would be on the new, unstable, dicentric chromosome. The acentric fragment that contains the left telomere of chromosome IX would be lost, and cell survival would depend on the retention of an unrearranged copy of chromosome IX. Depending on the secondary rearrangements of the dicentric chromosome, essential genes from chromosome II might be lost as well. Therefore, we expect that these cells might also retain an intact copy of chromosome II (Fig 2B and 2E). If an inverted circular intermediate integrated into chromosome IX to generate a Ura+ clone, the only change in karyotype would be an increase in the size of chromosome IX and a 2:1 ratio of the 5'*SUL1* probe relative to chromosome II (Fig 2C and 2E). Finally, if an inverted linear were to recombine with the *ra3* sequences on chromosome IX (Fig 2D), then a new chromosome would be created that has two copies of 5'*SUL1* sequence (also a 2:1 ratio of hybridization signal) and the right telomeres from chromosomes II and IX (Fig 2D and 2E). The reciprocal product, being acentric, would be lost and thus necessitate the retention of an unrearranged copy of chromosome IX.

Among the 27 Ura+ clones we examined in our initial experiment, five clones (18%) had a 2:1 ratio of the 5'*SUL1* sequences on the altered chromosome (Fig 3A). In each case, unrearranged copies of chromosomes II and IX remained, and the new chromosome had the centromere from chromosome IX and two copies of the *ura* sequence from chromosome II with variable amounts of centromere proximal DNA. These observations were consistent with an inverted linear ODIRA recombination (Fig 2D) in which the altered chromosomes were generated by recombination between a copy of chromosome IX and an inverted linear intermediate derived from chromosome II. To confirm the sequence arrangement of the new chromosomes in these five strains, we performed aCGH to determine the identity of the genetic material on the new chromosomes (an example is shown in Fig 3B). The most parsimonious assembly of the extra copies of segments from chromosomes II and IX (Fig 3C) is consistent with the new chromosome's size (Fig 3D). CHEF gel analysis and Southern hybridization followed by aCGH (S4 and S5 Tables) allowed us to determine the structure of the other 22 Ura+ clones, three of which are shown in S3, S4, and S5 Figs and summarized in S6 Fig. Each of the remaining 22 clones had structures that were consistent with the rearranged products of dicentrics that were produced by direct recombination at the *ra* sequences on chromosomes II and IX (Fig 2B) followed by centromere deletion (n = 7; example in S4 Fig), telomere capture either by BIR (n = 6) or de novo telomere addition (n = 6; example in S5 Fig), or breakage that initiated a bridge-breakage-fusion (BBF) cycle (n = 3; example in S6 Fig). We failed to recover any cases where chromosome II had the internal inverted triplication that would result from integration of a circular, inverted intermediate. However, the recovery of five clones with inverted chromosome II sequences appended to chromosome IX were consistent with the linear hairpin ODIRA model.

After establishing the patterns of ODIRA-related events via aCGH and Southern blotting, we screened an additional 23 Ura+ clones for ones in which the 5'*SUL1* probe on the new chromosome was in a ratio of 2:1 (relative to the signal on chromosome II). We recovered six additional clones for a total of 11 of 50 Ura+ clones. ArrayCGH analysis of each of these clones (S3 Fig) revealed the same pattern of copy number variation: 3 copies of chromosome II-right and

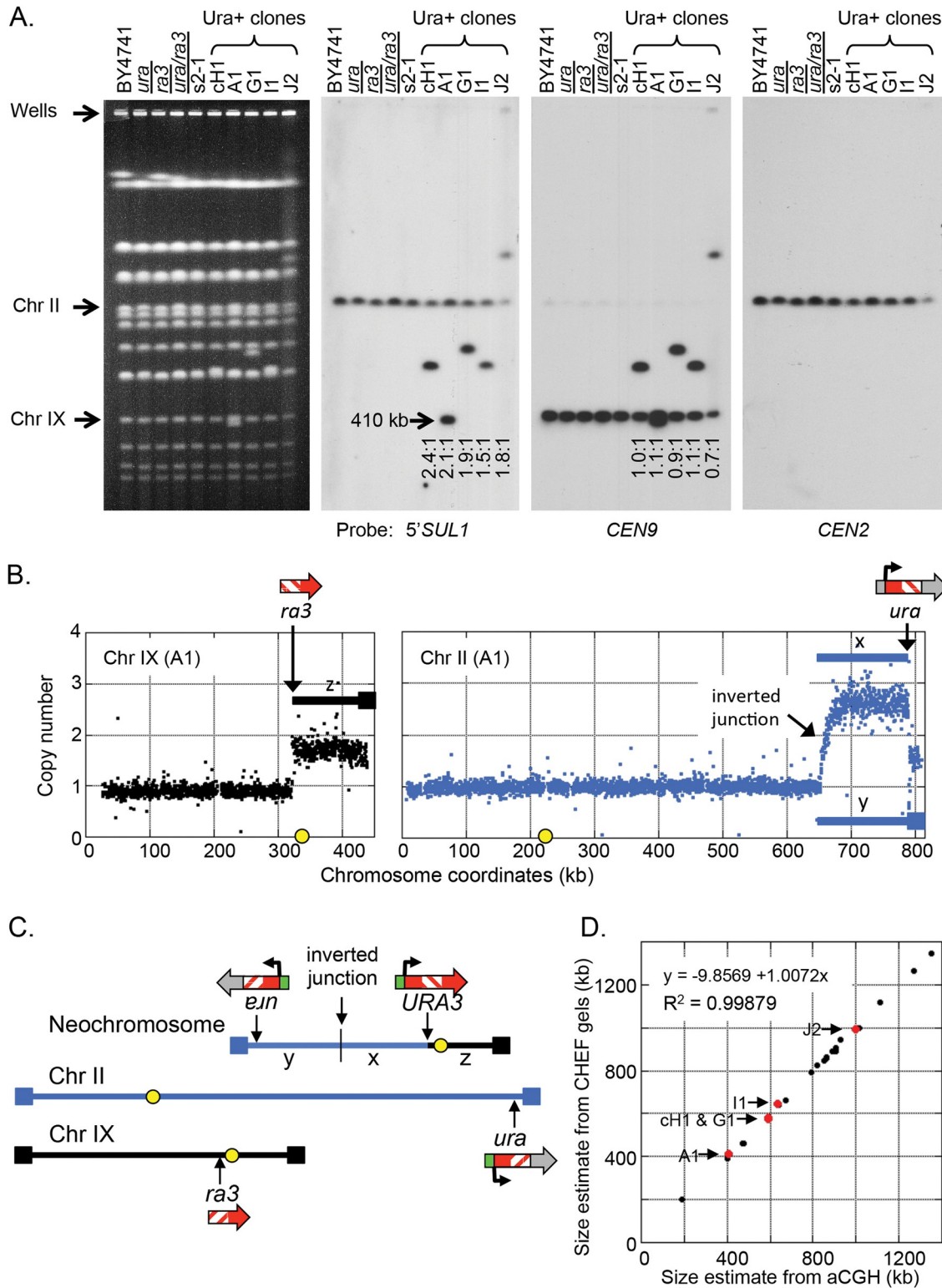

**Fig 3. Physical characterization of chromosomes from Ura+ clones.** (A) Chromosomes from a wild type strain, the haploid *ura* and *ra3* strains, the heterozygous diploid strain, split-*ura*3 strain, and five Ura+ clones were analyzed by CHEF gel electrophoresis and Southern blot hybridization. The ethidium bromide stained gel and the Southern hybridizations reveal that the Ura+ clones contain an additional single unique chromosome that hybridizes to the centromere of chromosome IX and also to the *5'SUL1* probe. The ratio of *5'SUL1* hybridization for each of the five Ura+ strains shows a ~2:1 hybridization ratio for the neochromosome relative to the

native chromosome II (measured by probe signal). (B) Array comparative genome hybridization of clone A1 vs. the parent split-*ura*3 strain indicates that the right telomeric segment of chromosome IX (labeled z) and the ~20 kb at the right telomere of chromosome II (as part of the fragment labeled y) are present at a copy number of ~2. A larger subtelomeric segment of chromosome II (labeled x) is present at a copy number of ~3. The sites of the *ura* and *ra3* insertions are indicated by arrows. (C) The proposed structure of the inverted neochromosome that contains the three amplified segments x, y, and z is illustrated above the two non-rearranged chromosomes. (D) The sizes of each of the five inverted neochromosomes (red dots) deduced from CHEF gels correspond to their predicted sizes based on aCGH data. Size estimates of non-inverted Ura+ chromosomes are indicated by black dots.

2 copies of chromosome IX-right, with discrete jumps in copy number at the *ura* sequence on chromosome II and the *ra3* sequence on chromosome IX. (This position marks the junction between chromosome II and chromosome IX, within the *ra* homology.) The junctions to the left of *SUL1* on chromosome II were at variable positions and did not show a discrete jump in copy number from one and three copies (S3 Fig). Rather, copy number gradually increased over an approximately 10 kb window, producing a pattern we refer to as a "waterfall" in which the copy number gradually transitioned from 1 to 3 copies. While many of the inversion junctions occur in close proximity to the *SUL1* locus some, such as J2 and G3A (S3 Fig), are at much greater distances.

The "waterfall" pattern of gradual copy number change is consistent with the inverted nature of the junction and the protocol we used for labeling the aCGH samples with Cy dyes (see methods; S2 Fig). Shearing of the genomic DNA before labeling removes the gradual transition in the aCGH profiles and converts the junction to a discontinuous one (S2 Fig). None of the other junctions attributed to dicentric rearrangements showed this gradual transition in copy number (see examples in S4, S5, and S6 Figs). In this final set of 50 Ura+ clones with the 2:1 5'*SUL1* hybridization ratio and the gradual copy number transition in the aCGH profiles, all were consistent with a recombination between an inverted linear acentric fragment derived from the *SUL1* region of chromosome II and the *ra3* locus on chromosome IX. If integration of a circular inverted intermediate occurs, it is below level of detection in our system, or is unique to the conditions in the sulfate-limiting chemostats.

## Reducing dNTP levels through inhibition of ribonucleotide reductase reduces the relative frequency of inverted Ura+ clones

Growing yeast cells in hydroxyurea reduces nucleotide pools [30] and alters features of the replication fork, including a ~16-fold reduction in fork speed [31], uncoupling of the replicative helicase (CMG) from the replisome [32], and an increase in the length of the single stranded gap [33]. The increased persistence and length of the single stranded regions increase the probability of single stranded breaks at forks, which would result in a single-ended double stranded break (Fig 4A). Because the broken dsDNA has no partner it cannot be repaired by end-joining mechanisms. However, the single-ended breaks are competent for BIR or homologous recombination and may be responsible for the increase in S-phase-specific homologous recombination events seen in HU treated cells [34]. In the split-*ura* strain such breaks at the telomere adjacent fork produce an end that can invade the homology on chromosome IX and generate the Ura+ clones through direct recombination or BIR. Single stranded breaks at the centromeric adjacent fork cannot produce Ura+ clones through these mechanisms. Nevertheless, end resection and fold over of breaks at the centromere-adjacent fork could produce a hairpin intermediate—the same intermediate we propose is produced by ODIRA (Fig 4A) that gives rise to Ura+ recombinants.

Growing the split-*ura* strain in the presence of hydroxyurea allows us to distinguish between ODIRA and single-ended double stranded break generation of Ura+ clones (Fig 2A). In the HU grown cultures, we recovered a roughly 3.5 to 5-fold higher frequency of Ura+ clones; however, examination of the clones revealed that 73 of the 76 clones were due to direct

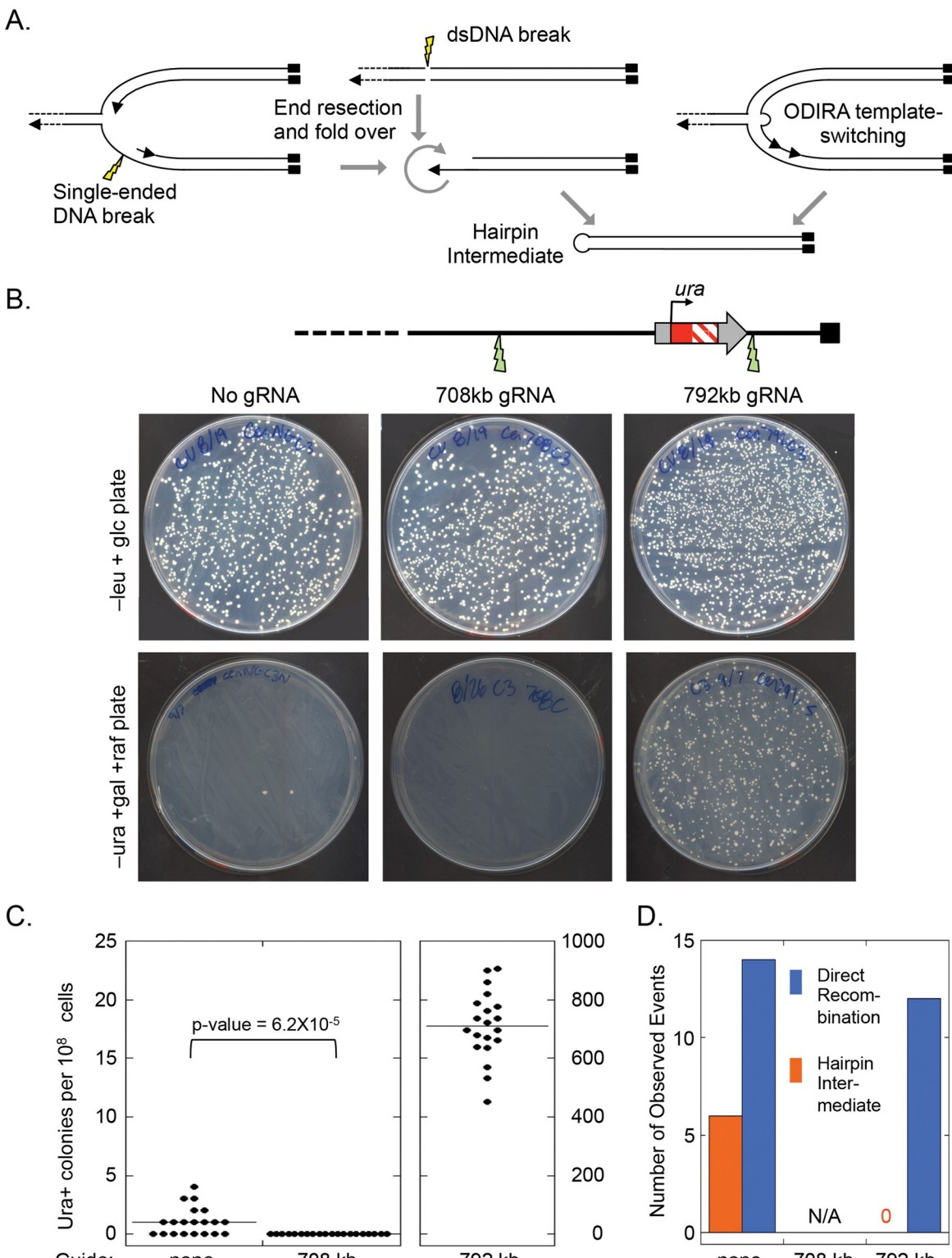

**Fig 4. Induction of DNA double stranded breaks by CRISPR/Cas9 cleavage centromere-proximal to *5'ura* (*SUL1*) eliminates inverted amplicons.** (A) The proposed hairpin intermediate could be formed (left to right) by a break in the single stranded DNA at a fork, by a double-stranded DNA break in nonreplicating DNA or by a single Cen-proximal ODIRA template-switching event. (B) CRISPR/Cas9 was used to induce a DSB either centromere-proximal (708 kb) or telomere-proximal (792 kb) to the *ura* locus. Cells that had been transformed with the CRISPR/Cas9 plasmid (S1 Fig) were selected for on –leucine plates. Twenty independent transformants

were grown to saturation in 1 ml of liquid –leucine medium and the entire saturated cultures (approximately $10^8$ cells) were spread on–uracil plates (-leucine, +galactose, +rafinose) to induce Cas9 expression and to select for successful regeneration of a functional *URA3* gene. (C) Cutting at 792 kb greatly increased Ura+ colony frequency while cutting at 708 resulted in no Ura+ colony recovery. (D) Cutting at 792 resulted in direct recombination events between chromosome II and IX, while cutting at 708 appeared to interfere with the ability to recreate the functional *URA3* gene. The control plasmid lacking a guide RNA gene produced a similar ratio of hairpin *URA3* chromosomes to recombined chromosomes (6:14) as found for the non-transformed split-*ura3* haploid strain (11:39).

recombination—the type produced by a telomere-proximal fork break. Only 3 Ura+ events (2 of 43 clones grown in 50 mM HU and 1 of 33 clones grown in 200 mM HU; clone HU++3 in S3 Fig was the single clone recovered from growth in 200 mM HU) were events that could be attributed to hairpin intermediates. If single-ended DNA breaks were responsible for the formation of inverted neochromosomes, we would have expected to see a similar 3.5 to 5-fold increase in the inverted events. These results suggest that inverted neochromosomes are not created by repair of a single-ended DNA intermediate.

## Centromere proximal double strand breaks do not generate inverted hairpin intermediates

To test whether a double-stranded break not associated with a replication fork could produce the hairpin intermediate (Fig 4A), we conducted CRISPR/Cas9 experiments on the split-*ura3* strain to ask specifically whether break repair is involved in the creation of the linear intermediates. If hairpins generated from rare spontaneous dsDNA breaks had contributed to the formation of the Ura+ clones we had observed in the split-*ura3* strain, then induction of dsDNA breaks centromere proximal to *SUL1* should increase the frequency of *URA3* products with an inverted structure (S8 Fig).

    We targeted CRISPR/Cas9 to a site centromere-proximal to *SUL1* (708 kb; Fig 4B). As a control, we also targeted a site distal (792 kb) to *SUL1* (Fig 4B). Cutting at this distal site should greatly increase the frequency of direct recombination events (Fig 2B) while reducing or eliminating inverted linear events (Fig 2D). To introduce these dsDNA breaks, we used *LEU2 CEN3* plasmids with *Cas9* under the control of the *GAL1* promoter (S1 Fig), selecting for transformants on –leucine plates supplemented with glucose to initially repress *Cas9* expression (Fig 4B top). We grew 20 individual colonies to saturation before plating on -uracil, -leucine, +galactose/raffinose plates to induce Cas9 and to select for uracil prototrophs (Fig 4B bottom). In the absence of a guide we recovered the typical one or two Ura+ colonies per plate (Fig 4B and 4C left) and among the independent Ura+ colonies we recovered six that produced the 2:1 *SUL1* hybridization ratio expected for linear ODIRA events (Fig 4D). As expected, cutting at the distal, 792 site resulted in a great increase in Ura+ colonies (~700-fold; Fig 4B and 4C right). While we only analyzed 20 of these Ura+ clones by CHEF gel hybridization, none were inverted products (Fig 4D) and all were consistent with repair of the dsDNA break by direct recombination with the *ra3* site on chromosome IX. In contrast, inducing cleavage at the proximal, 708 site completely eliminated the production of Ura+ colonies (Fig 4B and 4C middle). As there are abundant short inverted repeats immediately distal to the 708 site where a foldback could have occurred, the lack of Ura+ clones following CEN-proximal cleavage indicates that proximal breaks do not generate Ura+ colonies with an inverted chromosome architecture.

## Inverted junctions are commonly recovered after long-term growth in sulfate-limited chemostat cultures

The evidence we collected with the split-*ura3* strain argues that the major form of ODIRA amplification is through a linear hairpin intermediate. Therefore, we wanted to ask if the same

is true in sulfate-limited chemostats. In our previous analysis of *SUL1* amplicons we characterized clones obtained after ~250 generations of selective growth in low-sulfate medium using aCGH to look for copy number changes, CHEF gel analysis to look for changes in chromosome sizes and to determine the junctions of the amplified *SUL1* sequences, and Southern blots of restriction digests and snap-back assays to look for inverted DNA junctions [17]. Because we analyzed only a few clones from each chemostat, we were unlikely to identify the unstable intermediates or to accurately assess the range of variants that might be present in the culture. Indeed, the dogbone and hairpin intermediates are only expected to be present for at most the G2/M/G1 phases of the cell cycle after they were created. Because the intermediates have an origin of replication we anticipate that they get replicated into the inverted linears or the inverted dimeric circles in the following S phase. It would be impossible to capture the exact moment when such an intermediate arose and to find it through sequencing. To address these limitations, in this current work we collected culture samples from the last day of growth (~250 generations) for 31 independent sulfate-limited chemostats and performed Illumina 150 bp paired end DNA sequencing on the population samples. After mapping the reads back to the yeast genome we assessed read depth to identify amplified segments (S9 Fig). As a control, we subjected the same yeast strain to 32 independent glucose-limiting chemostats—where there is no selection for *SUL1* amplification—to assess the rate of false junctions created by PCR artifacts in the sequencing protocol.

Read depth analysis for the sulfate-limited cultures showed amplification of the *SUL1* gene and/or flanking sequences in all 31 populations (S9 Fig). The major amplification products appear to be interstitial; however, in five of the cultures we detected one or more amplification events that appeared to extend through *SUL1* to the telomere (S9 Fig, red arrow heads and squares). As expected, none of the 32 glucose-limited chemostats had amplifications of the *SUL1* region.

To examine amplification junctions, we searched for split reads flanking *SUL1* where one of the 150 bp reads mapped to two non-contiguous sites in the genome. There were three major types of junctions that could be confidently mapped: (1) junctions that map to two sites on the same strand of DNA indicating they are involved in and mark the sites of tandem duplications (S9 Fig, orange bars; chemostats S15, S19, and S23); (2) junctions between unique sequences and telomeric sequences that indicate that a terminal fragment of chromosome II had become appended to an existing telomere (S9 Fig, red squares; chemostats S11, S19, S21, S28 and S29); and (3) inverted junctions that mark the limits of inverted amplicons (S9 Fig, yellow circles; all chemostats with the exception of S19 and S28).

The vast majority of the split reads defined inverted junctions (Fig 5A and 5B); however, the depth of these reads varied widely. To determine a cutoff threshold for significance, we required that a junction be detected in two unique reads (with different ends) from the population being sampled. In part, this threshold was determined by the comparison to the glucose-limited culture split reads. While we found single reads from inverted junctions in the glucose-grown samples, there were no junctions supported by two independent reads. In addition, there was a similar frequency of single-read inverted junctions in the glucose- and sulfate-limited chemostats (S7 and S8 Tables). The second feature we used to set the two-read threshold takes into consideration the orientation of the split reads with respect to the *SUL1* locus. To be associated with inverted amplification of the *SUL1* region, the orientation of the split reads is important: in the productive orientation (Fig 5C) the centromere-proximal junction (CJ) is oriented with the duplicated segment extending toward *SUL1* and the telomeric end of the chromosome; likewise, the productive orientation for the telomere-proximal junction (TJ) is oriented with the duplicated region extending towards *SUL1* and the centromere. The same sequences could also be involved in non-productive orientations (Fig 5C) but these would not

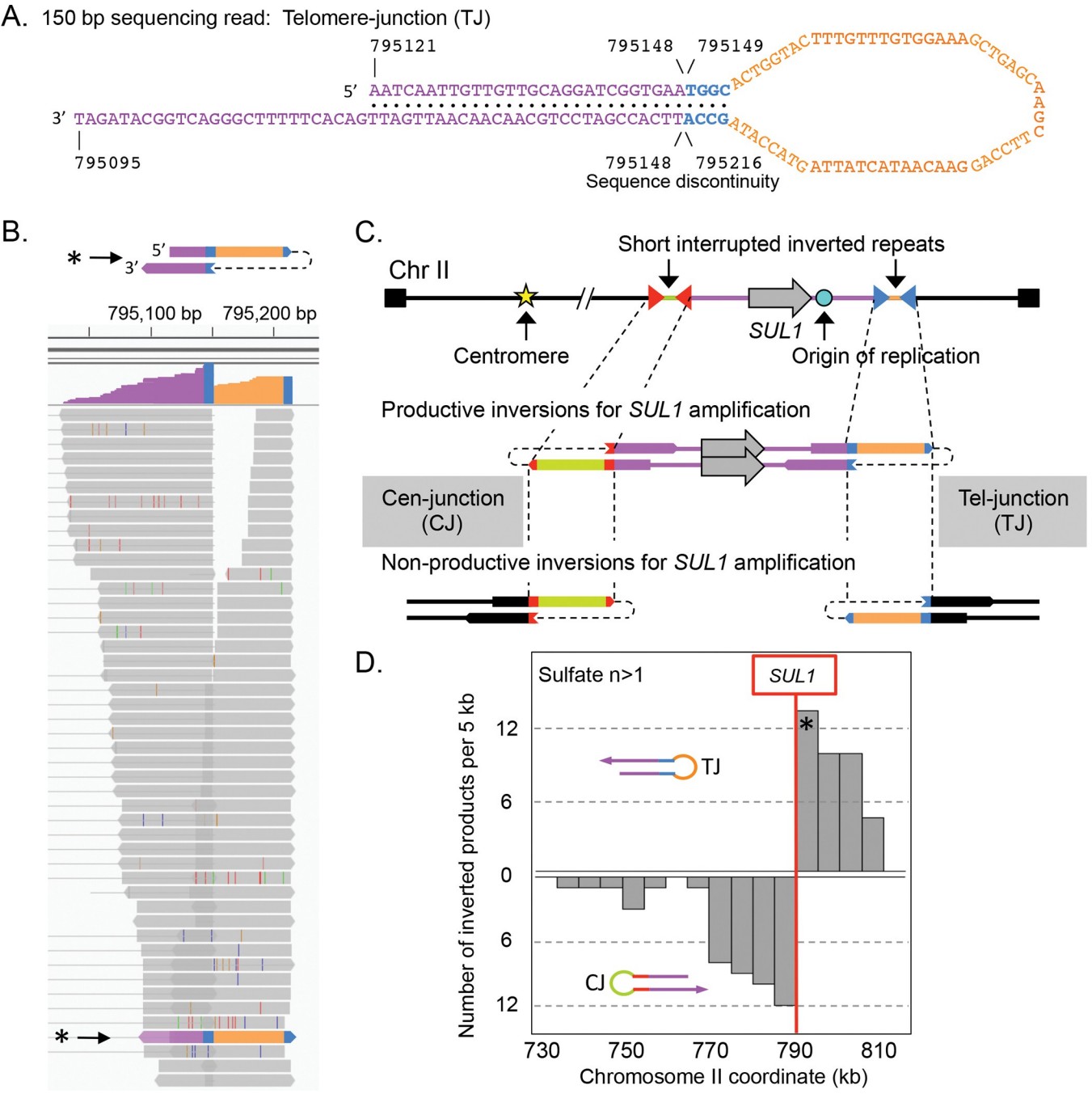

**Fig 5. Sequence analysis of inverted junctions from chemostat populations.** (A) An example of one split read from a population of cells grown ~250 generations in a sulfate-limited chemostat. In the 150 bp sequence the Watson strand rejoined the Crick strand at a four base pair inverted repeat (TGGC/GCCA). (B) The appearance of split reads in IGV. Each line is a single unique read that covers the same junction. The blue bars highlight the 4 bp inverted repeat, the orange segment is the 61 nucleotide-interruption that separates the two copies of the inverted repeats, and purple indicates the region that is present twice in inverted orientation in the 150 bp split read. The thin vertical colored bars indicate sequencing errors that differ from the reference genome. (C) The right end of Chromosome II with two short interrupted, inverted repeats (red and blue triangles) flanking *SUL1* and its adjacent origin *ARS228* are schematized. The inverted repeats could potentially create inverted junctions in two orientations. For the inverted junctions to be part of a *SUL1* amplification the Cen-proximal junction (CJ) must be oriented with the single stranded loop on the left and the Tel-proximal junction (TJ) must be oriented with the single stranded loop on the right. The other orientations, while involving the same interrupted inverted repeats, would not be productive for creating a *SUL1* amplicon. (D) Among the 31 sulfate-limited chemostat populations we catalogued 92 junctions that were represented by at least two independent sequence reads. The inverted junctions are distributed across the terminal ~80 kb of chromosome II with a perfect split in orientations that occurs to either side of the *SUL1* gene. The asterisks in (B) and (D) refer to the specific split read sequence shown in (A).

be associated with *SUL1* amplification. In a scan of all inverted junctions supported by sequence reads from two or more independent reads, there is a perfect concordance between their positions and orientations with respect to *SUL1* (Fig 5D). In contrast, no inverted junctions supported by two or more were found in the glucose-limited cultures and inverted junctions with a single read were oriented randomly with respect to *SUL1* (S10A Fig). In addition, comparisons of the sizes of the repeats and their spacing (S10B and S10C Fig) between split reads from the sulfate-limited populations that were found once and those that were found more than once show significant differences, validating our split-read threshold.

If inverted amplification occurs through sequential hairpin formation then we would expect to find cases where the numbers of centromere-proximal junctions do not match the number of telomere-proximal junctions (S11A and S11B Fig). Most populations contained multiple proximal and distal junctions; only three of the populations contained a single amplification event where the two oppositely oriented inverted junctions correspond with the edges of the amplified region (S9 Fig, chemostats S9, S25 and S27). For these populations a single expanded clone with a specific inverted triplication could have resulted from either insertion of an inverted circular ODIRA intermediate or by recombination with a doubly inverted linear (S11C Fig). We tabulated the distribution of inverted junctions across the 31 sulfate-limited populations (S6 Table) and found roughly a third of the populations had mismatched numbers of Cen- and Tel-proximal junctions (CJ and TJ; 11 of 31 cultures; S11D Fig). While we may have missed some junctions due to their rarity and limited sequence coverage, these results are consistent with the hypothesis that the left and right junctions do not occur simultaneously and favor the hairpin-ODIRA model proposed in Fig 1B.

## Genomic features at the sites of inverted junctions

All of the inversion junctions occurred at preexisting interrupted inverted repeats in the genome that ranged in size from 2 to 14 bp. We assessed the distribution, sizes, and spacing of interrupted inverted repeats ($\leq$250 bp) across the terminal 81 kb of chromosome II (Fig 6A and 6B). Potential inverted repeats are uniformly and surprisingly frequent with their frequency inversely correlated with the size of the repeats (Fig 6A and 6C). For the 31 sulfate-limited cultures, when we compare the repeats that were recovered in 89 of the 92 inverted junctions with their distribution and properties in this region of chromosome II (Fig 6C), we find that the longer the repeat, the more likely it will participate in an inversion event (Fig 6C and 6D). Spacing between repeats also determines which repeats result in inversion events (Fig 6B). Those with spacing between 40–80 bp are preferred while the genome distribution is relatively constant across all possible spacing intervals (Fig 6B). Both the length of the repeats and their spacing are similar to those found by Lauer et al. [35] among 28 inverted amplification junctions at the *DUR3* and *GAP1* loci and are consistent with the ODIRA mechanism.

Three out of the *SUL1* 92 inverted junctions identified—two CJ and one TJ—had spacing of inverted repeats of greater than 1 kb and had likely undergone a secondary deletion of one of the inverted arms. Since the initiating site of inversion could not be deduced, they were omitted from the above analysis. Similar secondary deletions were also detected in two of the split-*ura3* inverted junctions (I1 and CH1, S3 Fig), and in both an artificial dogbone construct [18] and in a previously characterized *SUL1* inverted triplication [18].

Given that optimal properties for frequency, size and spacing of repeats were uniformly distributed across the terminal 81 kb of chromosome II, it was surprising that the position of productive CJs were concentrated in the ~20 kb centromere-proximal to *SUL1* (Fig 6E). The clustering of CJs in the region between 770 and 790 kb (Fig 6E) cannot be explained by any bias in the distribution of potential sites but is likely the consequence of some other feature of

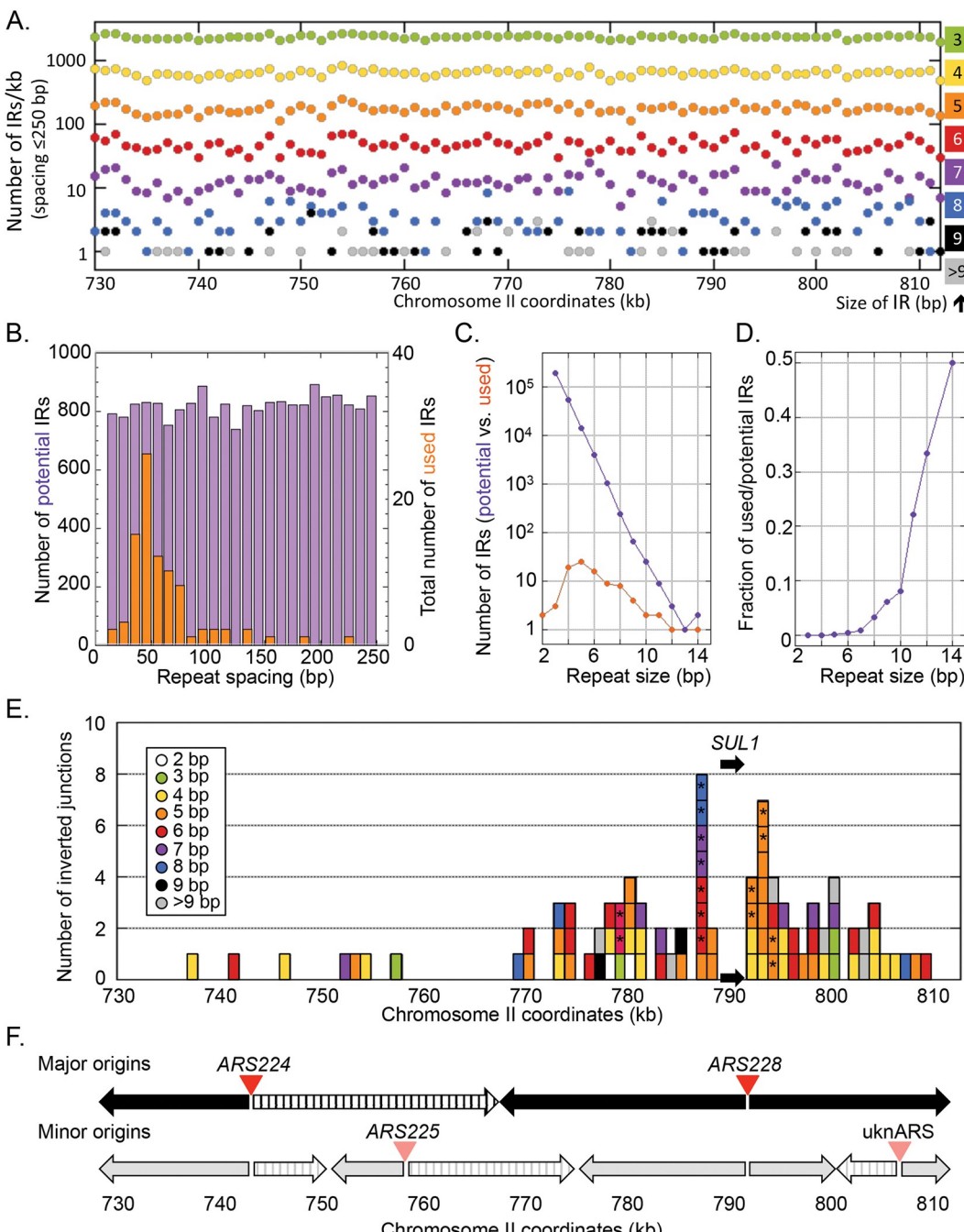

**Fig 6. Genomic features of the inverted junctions of *SUL1* amplicons.** (A) Density of genomic inverted repeat (IR) sequences (from 2 to 14 bp) that lie within 250 bp of each other across the 81 kb right terminal region of chromosome II, binned in 1 kb intervals. (B) The number of interrupted inverted repeats of different sizes (IRs + spacers in bp) in the terminal 81 kb of chromosome II (purple) is compared to the sizes of interrupted inverted repeats (orange) used to generate inverted *SUL1* amplicons. The number of inverted repeats of different sizes is relatively constant across the genome, but most of the repeats that give rise to inverted amplicons range from 40–80 bp. (C) The frequencies of inverted repeat sizes (2–14 bp) in the genome (purple) vs. those found at the sites of inverted amplicons (orange). (D) The ratio of repeats present in amplicon junctions (used) relative to their abundance in the genome (potential). (E) The location of each of 89 inverted junctions lying between 730 and 813 kb of chromosome II. The colors indicate the size of the repeats at the inverted junctions and are the same as illustrated in (A). The asterisks indicate that the same sequence was observed in 2 or 3 independent cultures. (F) Replication origins and the direction of fork movement in the 81 kb at the right end of chromosome II. Top: The most active origins (*ARS224* and *ARS228*) generate bidirectional forks moving left and right (arrows). The region between 743 and 768 kb is replicated by a rightward moving fork and therefore cannot create the Cen-junctions needed to generate

*SUL1* amplicons (striped arrow). Bottom: Infrequent initiation at two additional origins (*ARS225* and unnamed ARS— uknARS) contribute change the direction that sections of the chromosome are replicated. The minor leftward moving fork between 750 and 760 kb may be responsible for the four inverted junctions in the region shown in (E).

this chromosomal region. One possibility we considered is the presence of a gene or genes to the left of 770 kb that confer a selective disadvantage when included on the *SUL1* amplicon. However, the work of Sunshine et al. [36], who examined the fitness cost of chromosomal fragments that include variable extents of *SUL1*-adjacent DNA rules out this possibility: the first significant drop-off in fitness only occurs when DNA centromere proximal of 535 kb is included in the amplicon.

As our model is based on a potential replication error, we examined data from a high-resolution genome-wide replication study to understand this regional selection for inverted junctions [37]. There are six confirmed potential origins (ARSs) in this region of the genome (oriDB) but replication studies indicate that two are predominantly used late in S phase to complete replication of the *SUL1* region (*ARS224* and *ARS228*; Fig 6F top). Bidirectional replication initiation from these two origins predicts that the region between 743 and 768 kb would be replicated rightward by a fork from *ARS224* and, as a result, would be unable to generate the centromere-proximal fork error needed in our model to amplify the *SUL1* region (striped arrow; Fig 6F). Two additional origins (*ARS225* and an unnamed ARS, uknARS; Fig 6F bottom) contribute to a lesser extent to replication of the region. Initiation at *ARS225* reverses the direction of replication in the region between 752 and 758 kb and may be responsible for the four centromere-proximal junctions recovered in that region.

The discontinuous nature of replication on the lagging strand at replication forks may also explain the preferential recovery of inverted repeat spacing in *SUL1* amplicons. The size of the gap on the lagging strand template at a replication fork is dynamic as the leading strand advances. However, based on the size of Okazaki fragments in yeast (165 bp; [38]), we can extrapolate that the lagging strand gaps could range from a minimum of 0 to 165 bp (or more). We propose that the range of inverted repeat spacings found at inverted junctions (between 40 and 80 bp; Fig 6B) is a direct result of the size of Okazaki gaps on the lagging strand. If the repeats are close together (<40 bp) it is likely that the stalled fork would result in both copies of the repeat re-annealing between the parental strands, leaving the exposed repeats on the leading strand to anneal with one another forming a hairpin on the leading strand (Fig 7A). At the other extreme, if the copies of the repeat are more than 80 bp apart, the displaced copy of the inverted repeat on the leading strand would find its complement on the lagging strand template already occupied by an Okazaki fragment (Fig 7B). The ideal inverted repeat spacing results in one of the repeats being single stranded in both the leading and lagging strands and thus available for strand switching (Fig 7C). We suggest that the preferred size range of inverted repeat spacing that is found in inverted junctions, along with our results from the split-*ura3* strains grown in hydroxyurea, is an additional argument for a template switch between leading and lagging strands at the core of the ODIRA models.

## Discussion

We provide evidence that template switching between the leading and lagging strands at the same replication fork generates inverted amplicons through linear, inverted, extrachromosomal intermediates. These data lead us to augment our original ODIRA model by providing evidence that the two template switches could occur in subsequent S phases yet produce the same inverted triplication event. However, using a split-*ura3* yeast strain we find that a single template switch that generates an inverted linear intermediate from the *SUL1* region of the

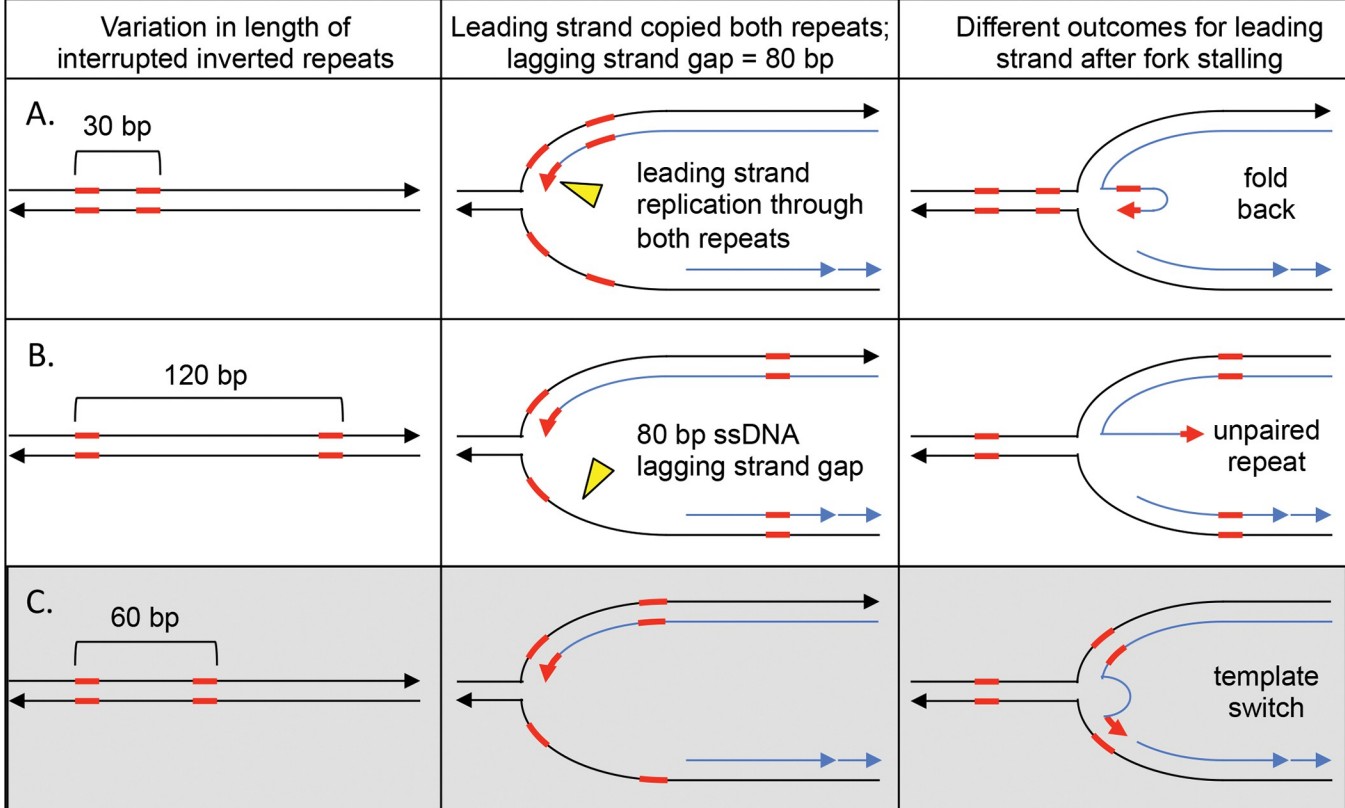

**Fig 7. The size of single stranded gaps on the lagging strand at replication forks limits which interrupted inverted repeats contribute to inversion junctions.** Interrupted inverted repeats with a spacing of 40–80 kb are overrepresented in the inverted junctions of *SUL1* amplicons. We propose that the size of the single stranded gap on the lagging strand is a primary contributor to this size selection. We illustrate this connection between interrupted inverted repeat size and lagging strand gap size for repeat spacing at three intervals. (A) If the repeats are in close proximity to one another (30 bp), during fork regression, the parental copies reform double stranded DNA and the displaced leading strand, containing the inverted repeats in single stranded form, are likely to self-hybridize. (B) If the repeats are at a distance that exceeds the average Okazaki gap on the lagging strand (120 bp), then fork regression leads to re-pairing of the parental strands of the left inverted repeat, while the right repeat had already been covered by an Okazaki fragment. This situation results in the repeat at the free 3'end with no template to which it can hybridize. (C) When both copies of the inverted repeats are within the size of an Okazaki gap (60 bp) then the left repeat will reform double stranded DNA between parental strands during fork regression. However, the left most copy on the leading strand still find its template in single stranded form on the lagging strand.

genome can arise in the absence of the sulfate-limited chemostat protocol and can be stabilized by selecting for its recombination with a second chromosome. In principle, the inverted linear intermediate is indistinguishable from one produced by fold-back repair of a double stranded break. However, inducing double strand breaks in the vicinity of *SUL1* does not promote the formation of inverted Ura+ clones. This finding, along with our previous work, leads us to conclude that a replication error is the initiating event in the formation of inverted amplicons.

A second line of evidence comes from split read analysis of populations of yeast grown in sulfate-limited chemostats where selective pressure often leads to cells with inverted triplications of the *SUL1* locus that sweep the population. Out of the 31 sulfate-limited populations we sequenced, 11 cultures had unmatched numbers of left and right inverted junctions. The spacing between the short inverted repeats that mark the centromere- and telomere-proximal junctions are compatible with the average size of Okazaki gaps on the lagging strand and provide the opportunity for the 3' end of the leading strand to hybridize with short complementary sequences on the lagging strand template. The distribution of centromere-proximal junctions is also consistent with the known direction of replication in the region centromere proximal to

*SUL1*. We should add that although we did not capture Ura+ clones that were produced through a dogbone intermediates in the split-*ura3* system, we cannot rule out the possibility that dogbone ODIRA occurs but at a lower frequency.

While we have not measured rates *per se*, the triplication of *SUL1* in sulfate limiting chemostats and the appearance of inverted Ura+ clones appear to be rare events. However, we do not know whether the template switching is rare, or whether the processing or recombination with the target chromosomal locus is limiting the recovery of the inverted products in both assays. For example, it is possible that template switching is not infrequent but rapid processing of the extruded hairpins or dogbones by structure specific nucleases interferes with the recovery of inverted products. The split-*ura3* yeast strain will make it feasible to look for suppressors and enhancers in known replication and repair pathway genes that participate in the generation of this unique form of CNV.

Reports of inverted CNVs in the human genome are increasing with the use of new long-range sequencing and optical mapping technologies. One common event is the DUP-TRP/INV-DUP configuration with the central copy in inverted order, flanked by direct duplication of shorter flanking sequences. This arrangement is easily explained by ODIRA, although the centromere- and telomere-proximal junctions do not have the same closely-spaced inverted repeat structure that we see in the majority of cases in yeast. Instead, these junctions can be explained by secondary rearrangements that increase the gap between the inverted arms of the inverted triplication (the DUP sequences). Indeed, in a survey of 45 representative localized CNVs with inverted segments in humans, we find that all such events can be explained by secondary rearrangements of an initial inverted triplication [39]. We have detected a few of these rearrangements in sulfate-limited chemostat cultures, the split-*ura3* yeast strain (this work) and in an artificial dogbone we introduced into yeast and then subjected to multiple passages in selective medium [18]. It is possible that conditions in the human germline or early development may be selecting for similar erosion of the palindromic arms of *de novo* events. Better characterization of inverted CNVs in different stages of cancer progression is needed and may provide us with insights into earlier stages of inverted CNV formation.

## Supporting information

**S1 Fig. Map of inducible CRISPR/Cas9 plasmid, pYCpGAL.**
(PDF)

**S2 Fig. The gradual copy number change in aCGH profiles is due to inverted junctions.** (A) aCGH of clone A1 shown for coordinates 500 to 813 kb. The DNA isolated for aCGH had an average size of ~20–40 kb. To label this DNA with Cy dyes, the DNA was denatured and random primers added for DNA polymerase to synthesize labeled strands. After labeling, the DNA was sonicated to ~500 bp and hybridized to an array. (B) The same DNA as in (A) was sonicated before denaturation and labeling. This method eliminated the gradual transition from 1 to 3 copies and produced an abrupt transition at the same site. (C) An illustration of the centromere proximal junction at ~650 kb of clone A1 with 50 bp of flanking genomic DNA. (D) DNA fragments of 20 kb are illustrated in a 3:1 proportion left and right of the centromere-proximal junction (CJ), respectively. (E) Representative molecules that either lack or contain the CJ palindrome. After denaturation and cooling the hairpins reform duplex DNA and are not available for priming and cy dye incorporation. Because breaks are in random places, different fragments will have different amounts of DNA excluded from the labeling reaction. These underrepresented regions are illustrated in gray in panel (D). (F) Quantification of the copy numbers expected across the site of the inversion junction generated from the schematic example in (D,E). Note that when the DNA was sheared to ~500 bp before labeling

(B), the region up to within 250 bp of the inverted junction was now available for labeling with cy dyes, with a resulting clean discontinuity in the aCGH signal in place of the previous water-fall pattern.
(PDF)

**S3 Fig. Array CGH of chromosome II derived from twelve Ura+ clones with 2:1 *5'SUL1* hybridization ratios.** All chromosomes were recovered after growth in normal medium with the exception of the clone labeled HU++3 that was obtained from incubation in 200 mM hydroxyurea. Notice the different features in the right and left amplification junctions. The left junction shows a gradual change in copy number from 1 to ~ 3 copies and occurs at variable sites along chromosome II centromere proximal to *SUL1*. The left boundary of each inversion junction shows the characteristic "waterfall" transition between copy numbers. In contrast, the right junction is an abrupt change in copy number from ~3 to 2 copies created by recombina-tion between *ura* and *ra3* (on chromosome IX). The aCGH profiles of chromosome IX were identical to that shown in Fig 2B left.
(PDF)

**S4 Fig. Centromere loss from a Ura+ dicentric chromosome produced by recombination between chromosomes II and IX.** (A) Ethidium bromide stained gel of clone C1 (center lane) with a neochromosome that is larger than the native chromosome II. (The flanking gel lanes contained other Ura+ isolates.) Southern hybridizations indicate that the neochromosome has retained *CEN9* and lost *CEN2* and that there is an unaltered version chromosome II. (B) ArrayCGH confirms that most of chromosome II has been duplicated but one copy of the chromosome has lost its *CEN2* sequence by recombination between the directly repeated Ty elements on either side of *CEN2*. The breaks in copy number on chromosome IX and the distal part of chromosome II mark the sites of the two *URA3* fragments. (C) The most parsimonious organization of the duplicated parts of chromosomes II and IX produce a neochromosome that is consistent with the size estimated from the CHEF gel.
(PDF)

**S5 Fig. Breakage and telomere addition stabilizes a Ura+ dicentric chromosome.** (A) Ethid-ium bromide stained gel of clone cH5 with a small neochromosome that retained *CEN9* (cen-ter lane; flanking lanes are from other Ura+ clones). The cells retained unrearranged chromosomes II and IX. (B) ArrayCGH confirms that the relevant part of chromosome IX has been duplicated, beginning at the insertion site of *ra3*; however only the right half of chromo-some II is duplicated, ending at the complementary region of the *ura* insertion. The sequence at ~420 kb could serve as a telomere seed after breakage of the dicentric chromosome. (C) The most parsimonious organization of the duplicated parts of chromosomes II and IX produce a neochromosome that is consistent with the size estimated from the CHEF gel.
(PDF)

**S6 Fig. Resolution of a dicentric Ura+ chromosome by a secondary breakage and recombi-nation event (McClintock Bridge-Breakage-Fusion, BBF).** (A) Ethidium bromide stained gel of clone cH5 with a very small neochromosome that hybridizes to *CEN9* and the 5'*SUL1* sequences from chromosome II. (B) ArrayCGH reveals that only a tiny fragment of chromo-some II is retained on this neochromosome and that the left telomere has been replaced by a second copy of chromosome IX right. (C) The most parsimonious organization of the dupli-cated parts of chromosomes II and IX produce a neochromosome that is consistent with the size estimated from the CHEF gel. Sequences of homology where the second copy of the right telomeric fragment of chromosome IX could have been added are shown in their relative

positions on the two native chromosomes.
(PDF)

**S7 Fig. Summary of structural rearrangements that gave rise to a cohort of 27 independent Ura+ clones.** (A) No integration of circular inverted intermediates was observed. (B) Five instances of recombination of an inverted linear with chromosome IX were obtained. (C) The remaining 22 events were produced by direct recombination between chromosome II and IX with subsequent loss of a centromere, breakage and addition of a telomere, or secondary recombination presumably as a result of breakage during mitosis through BBF cycles.
(PDF)

**S8 Fig. Alternative model for the generation of hairpin intermediates in *SUL1* amplification.** (A) Repair of a double stranded break could expose an inverted repeat in the 3'overhang that could be repaired to create the hairpin linear and its replicated isochromosomal fragment. If this double stranded break repair mechanism is responsible for the inverted Ura+ clones then CRISPR/Cas9 directed cutting on the Cen-proximal side of the *SUL1* region should lead to an increase in Ura+ clones overall and an increase frequency of inverted outcomes. (B) Resection of the 5'end of a double strand breaks introduced distal to *SUL1* would expose the <u>ura</u> homology in single stranded form and stimulate recombination with the *ra3* sequences on chromosome IX. CRISPR/Cas9 cleavage on the Tel-proximal side of *SUL1* should increase the overall frequency of Ura+ clones that occur through recombination events between the two chromosomes.
(PDF)

**S9 Fig. Read depth analysis of the right telomeric regions of chromosome II from 31 sulfate-limited populations.** 31 chemostat populations (S01-S31; ~250 generations) were subjected to 150 bp paired end Illumina sequencing. The read depth is shown as a heat map with higher copy numbers in darker shades of blue. The coordinates (in kb) are shown across the top X axis and the positions of ORFS (navy boxes) and several identified genes are shown across the bottom X axis. The position of *SUL1* is marked by the white dotted lines. The split reads that mark various types of amplification junctions are indicated by yellow circles (inverted junctions), orange bars (direct repeat junctions), and red squares (junction with an existing telomere—terminal translocations, indicated by the red triangles). Three of the inverted junctions (S06, S22 and S24) occurred in sequences to the left of 730 kb.
(PDF)

**S10 Fig. Comparison of split reads represented by a single PCR fragment to those represented by two or more unique PCR fragments.** (A) The distribution of inverted Cen- and Tel-junctions recovered from 31 sulfate-limited chemostats (top) and 32 glucose-limited chemostats (bottom) that were represented by single PCR fragment sequences (n = 1). No specific orientation with respect to *SUL1* was observed (compare to Fig 5D for split reads with support from two or more PCR fragments). Of note, there were no inverted junctions in the glucose-limited chemostat populations with support from two or more PCR fragments. (B) The size of the inverted repeats from the sulfate-limited chemostats with support from two or more PCR fragments (left) relative to those with support from one PCR fragment (right) are distinctly different. (C) The spacing between the inverted repeats from the sulfate-limited chemostats with support from two or more PCR fragments (left) relative to those with support from one PCR fragment (right) are distinctly different. These results provided the read-depth cut-off for distinguishing PCR artifacts from bone-fide *in vivo* inverted junctions. Data for all inverted junctions (n>1) in the sulfate-limited chemostats are in S6 Table. Data for all inverted junctions

(n = 1) in the sulfate- and glucose-limited chemostats are in S7 and S8 Tables, respectively. (PDF)

**S11 Fig. Imbalance in numbers of Cen-junctions and Tel-junctions in the 31 sulfate-limited chemostat cultures.** If inverted amplicons of the *SUL1* region arise through hairpin intermediates then in each population the number of Cen- and Tel-junctions would not necessarily be equal. (A and B) Mechanisms to explain unequal numbers of Cen- and Tel-junctions. Colored triangles indicate different inverted repeats along the chromosome. (C) Mechanisms to explain an equivalence between Cen- and Tel- junctions. (D) Among the 31 sulfate-limited chemostats, two had no inverted amplicons of the *SUL1* gene (one was a tandem duplication and the other was an amplification of the *SUL1* promoter region). The remaining 29 produced the total number of 92 inverted junctions. Eighteen cultures had matched numbers of Cen- and Tel-junctions. The remaining eleven had unbalanced numbers of Cen- and Tel-junctions. (PDF)

**S1 Table. Oligonucleotides used in strain construction.**
(XLSX)

**S2 Table. Primers for Southern probes.**
(XLSX)

**S3 Table. Guide RNA and oligonucleotide sequences for CRISPR/Cas9 plasmids.**
(XLSX)

**S4 Table. *SUL1* amplicon junction sequences (n>1 instances).**
(XLSX)

**S5 Table. Chromosome II arrayCGH data.**
(XLSX)

**S6 Table. Chromosome IX arrayCGH data.**
(XLSX)

**S7 Table. Sulfate-limited chemostats junction sequences (n = 1).**
(XLSX)

**S8 Table. Glucose-limited chemostats junctions sequences (n = 1).**
(XLSX)

**S9 Table. Numerical values used for graphs in Figures and Supplemental Figures.**
(XLSX)

## Author Contributions

**Conceptualization:** Claudia Y. Espinoza, Maitreya J. Dunham, M. K. Raghuraman, Bonita J. Brewer.

**Data curation:** Rebecca Martin, Christopher R. L. Large, Aaron W. Miller, Bonita J. Brewer.

**Formal analysis:** Rebecca Martin, Bonita J. Brewer.

**Funding acquisition:** Maitreya J. Dunham, M. K. Raghuraman, Bonita J. Brewer.

**Investigation:** Rebecca Martin, Claudia Y. Espinoza, Joshua Rosswork, Cole Van Bruinisse, Aaron W. Miller, Madison Miller, Samantha Paskvan, Gina M. Alvino, M. K. Raghuraman, Bonita J. Brewer.

**Methodology:** Rebecca Martin, Claudia Y. Espinoza, Aaron W. Miller, Maitreya J. Dunham, Bonita J. Brewer.

**Project administration:** Maitreya J. Dunham, M. K. Raghuraman, Bonita J. Brewer.

**Resources:** Rebecca Martin, Claudia Y. Espinoza, Aaron W. Miller, Joseph C. Sanchez, Maitreya J. Dunham, M. K. Raghuraman, Bonita J. Brewer.

**Supervision:** Rebecca Martin, Gina M. Alvino, Maitreya J. Dunham, M. K. Raghuraman, Bonita J. Brewer.

**Validation:** Rebecca Martin, Bonita J. Brewer.

**Visualization:** Rebecca Martin, Bonita J. Brewer.

**Writing – original draft:** Bonita J. Brewer.

**Writing – review & editing:** Rebecca Martin, Claudia Y. Espinoza, Maitreya J. Dunham, M. K. Raghuraman, Bonita J. Brewer.

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
