## [Decision Letter · Decision Letter 0]

19 Aug 2023

Dear Bonnie,

Thank you very much for submitting your Research Article entitled 'Template switching between the leading and lagging strands at replication forks generates inverted copy number variants through hairpin-capped extrachromosomal DNA' to PLOS Genetics.

The manuscript was fully evaluated at the editorial level and by independent peer reviewers. The reviewers were generally positive and appreciated the attention to an important topic but identified some concerns that we ask you address in a revised manuscript.

We therefore ask you to modify the manuscript according to the review recommendations. Your revisions should address the specific points made by each reviewer.

Yours sincerely,

Michael Lichten, Ph.D.

Academic Editor

PLOS Genetics

Gregory P. Copenhaver

Editor-in-Chief

PLOS Genetics

Reviewer's Responses to Questions

**Comments to the Authors:**

Reviewer #1: Martin et al investigate the mechanism underlying a specific class of copy number variation (CNV). Previously, this group has proposed a DNA replication-based mechanism, origin dependent inverted repeat amplification (ODIRA), to explain the occurrence of CNVs that are distinguished by a triplication with the internal repeat segment present in the inverse orientation and the flanking regions of the CNV containing short inverted repeat sequences. The proposed mechanism has been used to explain amplification at the SUL1 locus in yeast lineages that arise during the course of adaptation to sulfur limited chemostats. It has also been suggested that this mechanism underlies some CNVs at the GAP1 locus in yeast lineages that arise during glutamine limited chemostat selections. A similar class of CNV has been reported in the human genome making an understanding of this mechanism of CNV formation of potentially broad interest.

In this study, the authors engineered a yeast strain with the goal of distinguishing alternative variants of the ODIRA mechanisms and distinguishing them from recombination based mechanisms. A split URA3 gene was created with half the gene on chromosome IX and half the gene on chromosome II. The authors selected for ura+ clones that generated a functional URA3 and resolved the chromosomal structures using a combination of CHEF gels, southern blotting, and array comparative hybridization. Based on these analyses the authors propose that ODIRA occurs through an extrachromosomal linear intermediate that they term a hairpin rather than a circular intermediate as previously proposed in their model. To further test the role of DNA replication the authors treat the cells with hydroxyurea and introduce DNA breaks. Finally, the authors analyze additional populations of yeast cells selected in sulfur limited chemostats to characterize putative ODIRA events at the SUL1 locus.

This is an interesting study that contributes to our understanding of the proposed mechanism of CNV formation. In general, the experiments are well-performed and analyzed. Prior to publication, the authors should address the following:

-I had a hard time understanding Figure 1, which should be improved to clearly explain the difference between the two proposed intermediates. I suggest 1) labeling the telomeres, 2) clearly distinguishing between ssDNA and ssDNA with a consistent color or line type, 3) when the “hairpin” molecule is dsDNA it is no longer a hairpin and so should be drawn as linear molecule tp make this clear, 4) use a different symbol for centromeres (e.g. circles) and origins (e.g. a square) rather than just a different color.

-Similarly, I had a hard time understanding Figure 2. I suggest using a consistent color/symbol system (as with Figure 1). It would be helpful to clearly indicate what the final products are (e.g. with some shading or an outline) versus the intermediates to understand the rationale for the expected CHEF gel and southern results.

-I think that Figure 1 implies that the final product from a circular or linear intermediate is identical. Is that correct? If so, it provides an additional justification for using the split ura system and this could be made clearer in the text.

-As the hairpin model is drawn in figure 1, there are four copies on the linear intermediate. I would think this could result in either 4, 3, 2, or 1 copies when it recombines with the chromosome depending where the crossover occurs. Is this correct and if so are these variable number copies observed in sulfur limited selections?

-In the summary and abstract the authors make a connection to this class of variation in humans, but do not address the connection in the results or discussion. It would be worth addressing in the discussion how relevant the results are to understanding the mechanism in humans and/or its distinction from FoSTeS. Is it possible to undertake an analysis of inverted repeat frequency in the human genome and/or look at short read data for evidence of the breakpoints occurring at IR sequences?

- The reference to the human study in which a 2:1 SNP ratio was observed suggests that a useful experiment would be to perform sulfur limited chemostat selections using a hybrid strain (e.g. BYxRM). Have the authors tried this? Would the outcome of this approach be informative about the underlying mechanism?

- Given that chemostat selected populations are heterogeneous the presence of multiple CNV clones can confound the resolution of breakpoints. The authors appear to be successful in doing this. Populations adapted to glucose are used as a control, but I would think a comparison to sequenced clones in which only one breakpoint is identified would be a better comparison as it includes a positive and negative control.

-The lack of inverted sequences in the glucose-limited chemostat is surprising. Is it not possible that amplifications that include HXT6/7 can also be generated through ODIRA?

-The interpretation for the “waterfall” signal should be better explained in the results, perhaps with a cartoon to explain how this signal is consistent with an inverted junction. It is not intuitive to me.

- If addition of HU inhibits the ODIRA mechanism one would expect that the overall rate of ura+ clones is reduced. The authors state that the proportion of inverted clones is reduced, but is the overall rate reduced?

- On page 26 it is unclear if the distribution of IRs is for chr II or the whole genome as the authors refer to “the genome”.

-The role of a circular and linear intermediate could be tested by synthesizing this element and testing whether it generates the expected product. As written it is unclear whether this has been tried for the dogbone model (see page 26) and/or if it was attempted in this study for either intermediate.

- The authors’ study is focused on SUL1, but they neglect to point out that there is evidence that ODIRA underlies CNVs at the GAP1 and DUR3 locus (see Lauer et al., PLoS Biology 2018). Mentioning this could increase the relevance of this study.

- Some of the properties of ODIRA breakpoints at the SUL1 locus that are described have been reported in Lauer et al. 2018 - e.g. IR length and spacing (see figure 4). It would be interesting to compare the properties of SUL1 ODIRA breakpoints to these data to determine how generalizable they are.

Reviewer #2: The manuscript by Martin, Espinoza et al extends the work of the Brewer / Raghuraman group in the characterization of the ODIRA mechanism these authors proposed from their earlier analyses of amplification events spanning the SUL1 locus in S. cerevisiae. ODIRA is a somewhat unique pathway in that it is associated with DNA replication and leads to a hallmark triplication with characteristic orientation and short inverted repeats. The SUL1 locus contains the right combination of ingredients for this class of events to occur and then expand clonally and be detected among S. cerevisiae cells evolved under sulfate limiting conditions. Interestingly, the signature rearrangement configuration found in SUL1 triplications resembles certain (even more) complex disease-associate triplications in the human genome, suggesting possible universal consequences of ODIRA to genome instability across species. The Brewer / Raghuraman group has steadily refined their characterization of ODIRA, and in this manuscript they continue to advance their work in two valuable ways.

First, they set up a clever system for interrogating the structural nature of the proposed aberrant replication DNA unit that serves as the key intermediate in ODIRA triplications. The experimental system described here is particularly significant because it offers a substrate for triplications to re-integrate on Chr IX, outside of the SUL1 locus in Chr II. This movement provides an important example for how ODIRA can lead to structural variation even at distant recipient loci that lack the requisite replication origin flanked by pairs of short inverted repeats. Notably, the results of these experiments prompted the authors to revise their own original model in favor of a hairpin intermediate, replacing the circular ODIRA intermediate they had proposed earlier. Their embracement of a new data-driven revision of their original model was quite refreshing to read.

The second advancement in this manuscript is a thorough analysis of ODIRA junctions among cultures evolved in sulfate-limited chemostats. By analyzing bulk WGS reads from cultures (rather than purified clones), they were able to increase the number of junctions scored, and then used that information to infer mechanisms and propose novel ODIRA details.

Overall, this clearly written manuscript presents rigorously conducted work that further advances our understanding of the ODIRA mechanism. The manuscript does open many new questions, and leaves most unanswered. Several aspects of ODIRA are still uncharacterized, for example, the identity of the enzymes that must carry-out the various steps, and the direct isolation and identification of the hairpin DNA intermediate. Nevertheless, this work provides valuable new insight into one of the ways in which inappropriate DNA replication can drive the formation of certain classes of chromosomal alterations, and structural genomic variation in other broader contexts.

Below are major and minor suggestions that authors and editors should consider to improve the manuscript, ordered as they appeared in the manuscript text:

Major:

-The final paragraphs of the Introduction read very much like they belong in the Results section. I suggest keeping the rationale short in the Introduction, and expand on the details of the approach in the Results.

Line 462, and the whole HU experiment. This is my biggest comment and concern. I disagree with the interpretation of the HU results. An alternative (an in my view more likely) possibility is that replication stress caused by the HU treatment simply increases direct ura/ra3 recombination generating more dicentrics among the Ura+ clones analyzed (similarly to the case of the distal [792kb] CRISPR DSB). In this plausible scenario, the proportion of ODIRA Ura+ clones would be lower with HU exposure than without. Treating cells with gamma-rays or other global recombinogenic agent unrelated to replication stress would lead to the same outcome. The current HU interpretation can only be supported directly if the authors are able to determine and compare the absolute (not relative) number of inverted Ura+ clones with and without HU.

-The Conclusions section is limited to the findings of the work, which I guess is appropriate. However, the manuscript is lacking in terms of discussion of the findings in light of the broader genome instability field. I believe the conclusions should be folded within a proper Discussion section. Some of the suggested points for that expanded section include:

i. It would be useful to discuss the broader implications of ODIRA in other system, in light of the findings obtained in this study.

ii. Another point that could be addressed is something about the duration of the hairpin intermediate, or approaches that might be used to eventually capture it for analysis. How ephemerous is it? Can one ever hope to isolate it?

iii. Finally, consider points about a potential roles of sequence-specific endonucleases during the early steps in ODIRA. The step immediately following the template switch (Fig. 1A2) creates structures that might be recognized and cleaved by some of these enzymes. Would absence of those activities allow more of the strand switches to remain intact and thus progress toward mature hairpin intermediate formation? These could be in some way suppressors of ODIRA. Suggested looking at some of the work reviewed in PMID: 34624742.

Minor:

Line 24: “Inherited and de novo copy number variation…” I think this sentence should be revised as some de novo mutations are actually inherited (mutations formed de novo in a parent during gametogenesis).

Line 74: “and template switching during replication (FoSTeS and MMBIR)”. Note that the term “template switching” used here and other points in the text, has also been used extensively in association with BIR, particularly work from L. Symington’s lab (starting in 2007 with Smith et al PMID: 17410126). Given that this manuscript addresses genome rearrangements in yeast, I think it is important to specify precisely of what is meant by “template switching” here (eg. switching from leading strand template to lagging strand template).

Line 75: Some of reviews cited here (7-12) are already getting a bit old, and therefore not include important advancements in the field. For example, whenever discussing mechanisms that involve microhomologies, it is essential to include mention of relatively-recent advancements in MMEJ mediated by Pol Theta (even though it is absent in S. cerevisae). That specific topic can be referenced for example from PMID: 32302896.

Line 163: “Together these results suggest that double-stranded DNA breaks are not…”. I do not agree with this statement. It needs to be revised. Maybe DSBs are not needed to generate the extrachromosomal intermediate (dogbone or hairpin; Fig. 1A4 and 1B5), but a DSB in that intermediate or in the chromosomal integration locus would be needed to trigger the final step in ODIRA (HR-mediated re-integration of that intermediate into the chromosome).

Line 197: I think it would be important to give the ra3 and ura integrations specific allele names that can be included in the genotype of strain S2-1. As presented, the strains genotype give no indication of the presence of these repeat substrates. One possibility it to designate them as insertion downstream of the nearest gene (eg. YFG1::ra3 and YFG2::ura)(YFG, Your Favorite Gene), or something else to indicate they are in the strain.

Also related, why are ra3 and ura sometimes underlined in the manuscript? It was unclear to me what was the meaning to the undelining.

Line 341: “Because the left arm of chromosome IX has no centromere…” Sentence needs revision. By definition, no chromosome arm can have a centromere, otherwise it would not be an arm.

Line 403: The terms “circular” and “dogbone” are sometimes used interchangeably. It would be best to use a single term throughout, in which case, it would be “dogbone” because it is more unique and catchy – readers will know exactly what it is about.

Reviewer #3: The focus of the manuscript from Dr. Brewer and her group is the new mechanism called ODIRA generating segmental inverted triplications similar to those frequently associated with human developmental and cancerous phenotypes. The idea of the elegant new mechanism proposed by the authors is that following replication pausing and fork regression, template switching occurs at inverted repeats followed by ligation of the leading and lagging strands. The authors propose that this event can generate an extrachromosomal intermediate that can later integrate into the chromosome generating segmental inverted triplications. The authors proposed two different variations of their model, built an experimental system in yeast to test these two variations of their ODIRA model, and successfully tested their predictions using their experimental system. The results of their test support the existence of one of their proposed mechanisms, hairpin ODIRA, but not another one, dogbone ODIRA. Based on their results, the authors also propose that replication stalls (and not DSBs) serve as the main inducers of ODIRA-events. In addition, the authors report the optimal parameters of short inverted DNA repeats mediating ODIRA. Together, the authors conclude that their data support a two-step version of ODIRA model, where sequential template switching events at short inverted repeats between leading and lagging strands at replication forks generate inverted interstitial triplications. The results presented in this elegant and detailed manuscript will be of great interest for the diverse readership of PLoS Genetics including researchers studying the mechanisms of chromosomal rearrangements including those leading to congenital diseases, cancer and to neurological syndromes.

Specific comments:

1. How did the authors decide on the position for the insertion of the “a3 piece”? I mean is there any reason for inserting it into the Chr IX and at this specific location at this chromosome?

2. P. 21, line 462-464. Following the exposure to HU, the authors observed the decrease of the fraction of inverted Ura+ clones among all selected Ura+ events. This led them to conclude that ODIRA is decreased in the presence of HU. What if HU exposure led to the increase of the competing events leading to Ura+ instead? For example, could the result be explained by HU stimulating chromosome breakage promoting direct recombination between Chromosomes II and IX producing Ura+?

3. Fig. 1B and the text of the paper: It is unclear how the extrachromosomal piece (the very bottom, right) ended up with two pairs of the purple inverted repeats.

4. Could the authors explain in detail the formation of the outcome J2 (Figure 3A). The neochromosome in this outcome appears very large for the product of dicentric, and it is not clear how it could be formed via the proposed ODIRA mechanism. At least some information on the size of the product and possible mechanism would help.

5. Fig. 3C: it would help to indicate the specific location/chromosome region hybridizing to

the probe “x”.

6. The data presented in the Figure 5B are not clear. More explanation is needed to this panel in general. Also, some specific details need to be explained. For example, what is depicted as thin stripes of different colors?

7. Fig. 5D: not clear what is the meaning of asterisks.

8. Fig. 6D: the legend mentions a striped arrow, but it is unclear which one. I did not find it.

9. Fig. S3 and the text: the authors compare the left and the right boundaries of the amplified region and indicate that the right is more abrupt than the left one. It is hard to tell. How rigorous this comparison was? It seems like there is a lot of variation.

10. Figure 7C, the last, right, column: is it right that the loop is formed in the blue (newly synthesized) strand? My impression is that the black strand will need to loop out.

11. General Discussion: is the presence of the origin next to the amplified region required for the formation of ODIRA-like events? The model requires the presence of the origin, and the events are called “origin-dependent”. However, is it possible that even without replication the fragment can still integrate and lead to amplification (for example as STR-type of event)?

**Have all data underlying the figures and results presented in the manuscript been provided?**

Reviewer #1: Yes

Reviewer #2: Yes

Reviewer #3: Yes

PLOS authors have the option to publish the peer review history of their article (what does this mean?). If published, this will include your full peer review and any attached files.

Reviewer #1: **Yes: **David Gresham

Reviewer #2: No

Reviewer #3: No

---

## [Decision Letter · Decision Letter 1]

23 Oct 2023

Dear Dr Brewer,

We are pleased to inform you that your manuscript entitled "Template switching between the leading and lagging strands at replication forks generates inverted copy number variants through hairpin-capped extrachromosomal DNA" has been editorially accepted for publication in PLOS Genetics. Congratulations!

Before your submission can be formally accepted and sent to production you will need to complete our formatting changes, which you will receive in a follow up email. If you wish to make any other changes, please contact the editorial office. Please be aware that it may take several days for you to receive this email; during this time no action is required by you. Please note: the accept date on your published article will reflect the date of this provisional acceptance, but your manuscript will not be scheduled for publication until the required changes have been made.

Yours sincerely,

Michael Lichten, Ph.D.

Academic Editor

PLOS Genetics

Gregory P. Copenhaver

Editor-in-Chief

PLOS Genetics

Comments from the reviewers (if applicable):

Reviewer's Responses to Questions

**Comments to the Authors:**

Reviewer #1: The authors have done an excellent job in responding to the critiques of the initial version of the manuscript and the paper is much improved. The authors might like to consider the following minor points in finalizing this important paper for publication.

1/ In Figure 1 I would find it most useful to draw the final product as a single linear molecule - this would make the inverted nature of the middle copy clearer and I am not sure why the structure needs to be draw with the copies aligned.

2/ The last sentence of the abstract should state; “ a two-step process…at a replication fork followed by integration through homologous recombination…”

Reviewer #2: The authors have adequately addressed my comments and concerns.

Reviewer #3: The authors addressed all of the reviewer's comments really well, and this further improved this manuscript that has been interesting and very impressive from the beginning.

**Have all data underlying the figures and results presented in the manuscript been provided?**

Reviewer #1: None

Reviewer #2: Yes

Reviewer #3: Yes

PLOS authors have the option to publish the peer review history of their article (what does this mean?). If published, this will include your full peer review and any attached files.

Reviewer #1: No

Reviewer #2: No

Reviewer #3: **Yes: **Anna Malkova

**Data Deposition**

http://datadryad.org/submit?journalID=pgenetics&manu=PGENETICS-D-23-00733R1

**Press Queries**

---

## [Editor Report · Acceptance letter]

11 Dec 2023

PGENETICS-D-23-00733R1 

Template switching between the leading and lagging strands at replication forks generates inverted copy number variants through hairpin-capped extrachromosomal DNA 

Dear Dr Brewer, 

We are pleased to inform you that your manuscript entitled "Template switching between the leading and lagging strands at replication forks generates inverted copy number variants through hairpin-capped extrachromosomal DNA" has been formally accepted for publication in PLOS Genetics! Your manuscript is now with our production department and you will be notified of the publication date in due course.

With kind regards,

Anita Estes

PLOS Genetics

On behalf of:
